# Group membership biases children's evaluation of evidence

**Joshua A. Confer** [1] ✉, **Allison M. Champ**[1], **Dorsa Amir**[2], **Hanna Schleihauf** [3,4,5] & **Jan M. Engelmann**[1,5]

People form beliefs not only as individual agents, but as members of social groups. Here, we investigate how group membership influences belief formation and revision in childhood. Across three studies ($N = 262$), 4–6-year-old children either joined one of two groups or neither group, then evaluated evidence to arrive at a conclusion. Children who belonged to a group were more convinced by evidence that supported their ingroup's belief (Study 1 & 2) and were less convinced by evidence that opposed their ingroup's belief (Study 3), leading them to hold inaccurate group beliefs. Children who did not belong to a group rationally evaluated the available evidence and arrived at accurate conclusions. These results suggest that group membership modulates children's evidentiary standards.

Why do people hold inaccurate beliefs in the face of overwhelming contradictory evidence? A central reason may be that we form beliefs not just as individuals, but also as social beings, embedded within groups[1]. We often know what other group members believe, and they, in turn, know what we believe. Theorists have argued that, compared to individual settings, group contexts introduce at least two additional influences on our beliefs. First, we often treat ingroup members as reliable sources and update our beliefs in light of their testimony[2–5]. Second, we feel pressure to align our beliefs with our groups to gain a sense of belongingness and avoid ostracism[1,6,7]. Such group influences can result in inaccurate beliefs when we trust or affiliate with the wrong source.

In the current project, we examine the developing psychological processes underlying belief formation and revision in group settings. Are young children, who are beginning to form core, lasting beliefs about the world[8,9], already influenced by their social groups? If so, how do children adjust their belief formation practices when they reason as a member of a social group? Answers to these questions have the potential to guide efforts across multiple disciplines—from education and social policy to public health—aimed at reducing polarization and fostering intellectual humility. Interventions targeted early in life may prevent the entrenchment of mature partisan biases, which can result in beliefs resistant to counterevidence[10–12].

Previous research on how children evaluate evidence to form beliefs has largely taken place in individual contexts, in which participating children adopt beliefs on their own, with no social influences or other agents involved. This line of research indicates that young children are akin to "little scientists" who are rational, open-minded, flexible, and curious learners[13–19]. For example, already 2-year-olds form beliefs based on a careful assessment of the available evidence[20–25]. At older ages, 4–5-year-olds become more sophisticated reasoners, who rationally revise their initially held beliefs if they later receive contradictory evidence[26–28], seek out evidence that goes against their beliefs[29–32], and search for more evidence when the evidence they have is inconclusive[33–36]. Taken together, this work suggests young children have a strong motivation to form accurate beliefs and use ideal epistemic practices to do so—at least in individual contexts.

In the real world, children, like adults, often reason about evidence and form beliefs as members of social groups. In these contexts, children may use others' testimony to guide their belief formation practices, treating others' beliefs as a source of evidence. Indeed, research has shown that even young children tend to accept a claim supported by an accurate source over an inaccurate source[37–39], a source who provides strong over weak reasons[28,40], and a unanimous majority over a lone dissenter[41]. However, children do not blindly trust any majority. Preschool-age children will give an obviously incorrect answer in public due to social pressure, yet, when asked privately by an experimenter, children demonstrate the correct answer, indicating

[1]Department of Psychology, University of California Berkeley, Berkeley, CA, USA. [2]Department of Psychology & Neuroscience, Duke University, Durham, NC, USA. [3]Social and Behavioural Sciences, Utrecht University, Utrecht, CS, Netherlands. [4]German Primate Center, Leibniz Institute for Primate Research, Göttingen, Germany. [5]These authors jointly supervised this work: Hanna Schleihauf, Jan M. Engelmann. ✉e-mail: confer@berkeley.edu

children's belief formation was unaffected[42,43]. This suggests that from an early age, children use other people's testimony to inform their beliefs and that they have sophisticated intuitions about who they should trust. However, we know little about how belonging to a group influences children's epistemic practices beyond testimony, including how they seek out and evaluate direct, first-hand evidence. Investigating children's epistemic practices within a group context can offer further insights into the early-emerging cognitive processes that underlie belief formation in the real world.

Examining the development of belief formation in group settings is an especially relevant test case, as there are strong theoretical reasons to expect that group membership introduces powerful social influences. As mentioned above, the groups we belong to may be viewed as reliable sources we can trust when forming our beliefs[44]. Empirical research has shown that already young children preferentially trust ingroup members when learning a word or listening to a story[45,46]. Additionally, our beliefs can also serve as markers of group membership, signaling to our group that we are one of them[1,7,47]. Under certain conditions, it may be more beneficial to hold an inaccurate belief and reap the social benefits, than to hold an accurate belief and face the social consequences.

Concern with group membership emerges early. Children as young as age 4 express a stronger liking of ingroup members, attribute more positive attributes to ingroup members, and share more resources with ingroup relative to outgroup members—even if these groups are only minimally marked[48–56]. Likewise, young children prefer to hear information from their ingroup[57,58]. Children at this age do not just prefer their ingroups, they also actively care about belonging to them and are finely attuned to the norms and expectations of doing so. For example, young children are concerned about their reputations, in particular what their ingroup members think of them[59–61]. This sensitivity helps ensure children act in ways to maintain and improve their reputation with their groups and avoid being excluded. Indeed, children make sophisticated judgments about who should be given group membership status and recognize the consequences of social exclusion[62–68], suggesting that they are acutely aware of the dynamic nature of group belonging.

Children also have specific expectations for what it means to be a member of a group, both in terms of how group members should behave and what they should believe. Children expect group members to perform similar behaviors as their group[69–74]. For instance, from the age of 4 onwards, children are more disapproving of group members who ate different food or played a different game than what was typical of the group relative to group members who behaved the same as the group[72]. Most relevant to the current study, children at this age also expect individuals to hold the same beliefs as their group in third-party contexts. Roberts et al. show that 4–6-year-olds are more disapproving of a group member who believed that a red ball was red when the group believed it was blue, relative to a scenario where the group member and the group both believed it was red[75]. Similarly, children think that group members should hold the same opinions and ideology-based beliefs as their group. Older children aged 7–9 possess even stronger expectations that others should hold group beliefs. These results indicate that from an early age, children want to belong to social groups and understand that part of being a group member is to hold group beliefs.

What is less understood is if children's growing concern with their group identities at this age also influences how children themselves evaluate evidence and form beliefs. Work with adults suggests that their epistemic practices are influenced by group membership and demonstrate a variety of ingroup biases: adults seek out group-supporting information, avoid opposing views, and place extra weight on information coming from their ingroups[7,47,76–81]. These patterns raise important questions about the trajectories and mechanisms through which beliefs are formed. What are the developmental roots

of these biases on evidence evaluation and belief formation? Does belonging to a group adjust young children's standards of evidence for holding group beliefs? Or are young children's own epistemic practices resistant to group biases?

Here, we investigated the ways in which group membership may influence children's evidentiary standards. Across three pre-registered studies, we tested whether belonging to a group influences young children's epistemic practices. In each study, children either belonged to one of two groups (in the Group Condition) or simply learned about two groups (in the No Group Condition). In Studies 1 and 2, we focused on belief formation. We tested the hypothesis that group membership lowers children's standards of evidence for forming group beliefs: do children require fewer pieces of ingroup-supporting evidence to form an ingroup belief? In Study 3, we focused on belief revision. We tested the hypothesis that group membership raises children's standards of evidence for revising their group beliefs: do children require more pieces of opposing evidence to revise an ingroup belief?

Taken together, the results of the three studies show that group membership biases preschool-aged children's epistemic practices. In Study 1, children as young as four sample less evidence when the initial evidence supports their group's belief. As a result, most children in the Group Condition hold an incorrect belief, following their group's belief. In Study 2, children who belong to a group are more convinced by evidence that supports a group belief than evidence that supports a non-affiliated group's belief. In Study 3, children are less convinced by evidence that opposes a group belief than evidence that opposes a non-affiliated group's belief. These findings indicate that already young children's standards of evidence vary as a function of social context.

## Results

### Study 1: How children sample evidence in group contexts

In Study 1, we examined whether 4–6-year-olds exhibit bias in how they seek out evidence that supports a group belief. Children first either joined one of two groups (in the Group Condition) or joined neither group and were simply told that there were two groups (in the No Group Condition). The primary task in our studies was a child-friendly reasoning game in which children had to determine, by evaluating the relevant evidence, whether there were more elephants or lions in a set of ten boxes on a table. Children were told that the two groups held different beliefs about the boxes. After hearing both beliefs, children could then gather evidence themselves by opening one box at a time, with a 30 s time delay in between each box. Importantly, the ten boxes were arranged such that the first five boxes children could acquire seemed to support the ingroup belief (for children in the Group Condition) or group 1's belief (for children in the No Group Condition). However, if examined in total, the evidence in fact confirmed the other group's belief in both conditions, four boxes to six (see Fig. 1B). Our two main dependent variables were: (i) how many boxes children opened and (ii) whether children held the correct or incorrect belief about the boxes. We hypothesized that children who belonged to a group would seek fewer pieces of evidence and more often form the incorrect group belief. We also predicted that this tendency would be stronger for older children than younger children (following the developmental trend in ref. 75).

We first analyzed whether group membership influenced children's decision to evaluate evidence through opening the boxes. We fit a linear model to predict the number of boxes opened (1–10, numeric), by condition (No Group, Group, factor) and age (4–6, continuous numeric), and their interaction as fixed effects. To avoid an increased type I error risk due to multiple testing, we first tested the overall effect of the predictors. Therefore, we compared the model fit of the full model to a null model including only the intercept. Then, to determine the effects of each predictor, we compared the full models to reduced models lacking the predictor of interest. All tests in Studies 1–3 were

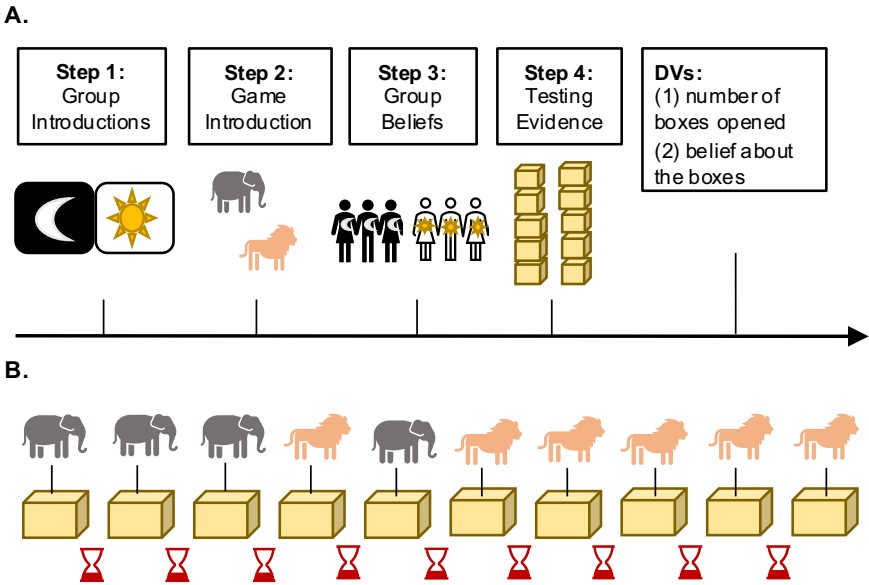

**Fig. 1 | Procedure and evidence order used in Study 1. A** The four-step procedure used in both conditions: Step 1: Children either joined one of two groups or were simply introduced to them. Step 2: Children were tasked with figuring out whether there were more elephants or lions in a set of ten boxes. Step 3: Children were told about the two groups' contrasting beliefs about the boxes. Step 4: Children could then open the boxes themselves until they arrived at a conclusion. **B** Order of evidence in Step 4: Children could open one box at a time and had to wait 30 s before opening the next box. The beginning boxes contained mostly elephants, however, in total, there were more lions.

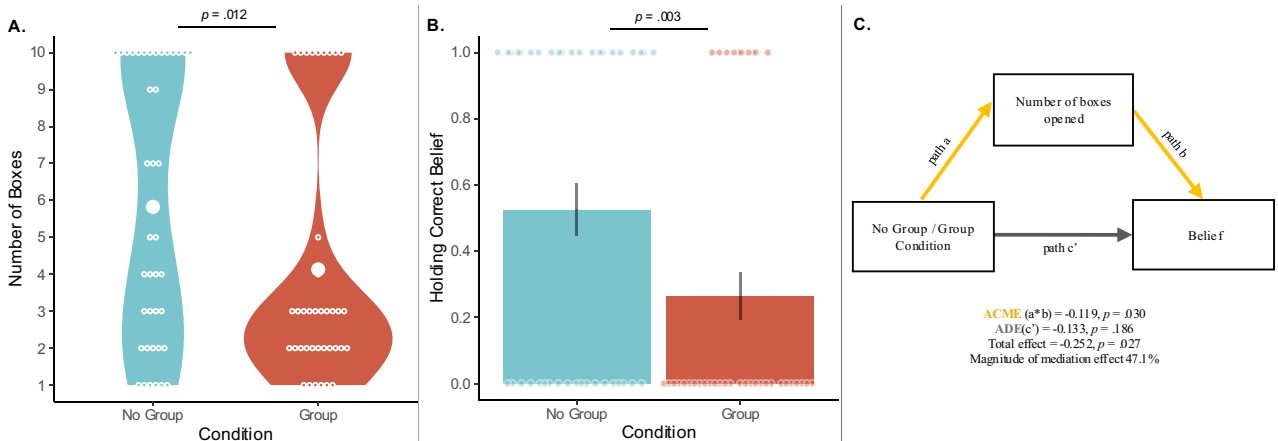

**Fig. 2 | Group influence on children's evidence-seeking and belief formation.**
**A** Number of boxes children opened by condition: The distribution of responses for children in the No Group Condition is shown in blue, and the distribution of responses for children in the Group Condition is shown in red. Individual data points (open circles) and condition means (closed circles) are plotted in white. Statistical test: model comparison testing the effect of condition as described in the main text (two-sided test; $\chi^2(1) = 78.15$, $p = 0.012$, $R^2 = 0.08$; $n = 78$ biologically independent replicates (children)). To account for multiple comparisons across panels, we first conducted a full-null model comparison before testing individual effects. **B** Proportion of children correctly believing there were more lions in the boxes: Data are presented as mean values +/− SEM, $n = 78$. Statistical test: model comparison testing the effect of condition as described in the main text (two-sided test; $\chi^2(1) = 8.53$, $p = 0.003$, $R^2 = 0.14$; $n = 78$). **C** Mediation analysis: The influence of condition on children's beliefs was primarily explained through the indirect effect of the mediator—the number of boxes they opened (represented by yellow arrows; two-sided tests). The average mediation effect (ACME) was significant: −0.119, 95% CI [−0.238, −0.011], $p = 0.030$. The average proportion mediated was 47.1%, 95% CI [0.004, 1.740], $p = 0.049$. The total effect of condition on belief was significant: −0.252, 95% CI [−0.460, −0.030], $p = 0.027$. The direct effect (ADE, c') was not significant: −0.133, 95% CI [−0.328, 0.063], $p = 0.186$.

two-sided. Statistical models were checked for violations of key assumptions, including residual normality and homoscedasticity; when assumptions were not met, we confirmed the robustness of results using non-parametric tests or GLMMs as confirmatory analyses (see Studies 1–3 under Supplementary Methods in the Supplementary Information for full diagnostic output). Analyses were conducted in R (version 4.5.1).

Children in the Group Condition opened significantly fewer boxes than children in the No Group Condition (see Fig. 2A). The full model

for boxes containing condition, age, and their interaction as predictors was a significantly better fit than the null model ($\chi^2(3) = 131.93$, $p = 0.012$, $R^2 = 0.13$). Next, we compared the full model to reduced models. We found a significant effect of condition ($\chi^2(1) = 78.15$, $p = 0.012$, $R^2 = 0.08$), such that how many boxes children opened (1–10) depended on whether they were in the Group Condition or No Group Condition. On average, children in the Group Condition opened about four boxes while children in the No Group Condition opened about six boxes. There was no statistically significant interaction between

condition and age ($\chi^2(1) = 38.10$, $p = 0.070$, $R^2 = 0.04$), nor was there a statistically significant effect of age ($\chi^2(1) = 37.94$, $p = 0.076$, $R^2 = 0.04$).

For our second dependent variable—children's beliefs about the boxes—we fit a generalized linear model, as belief was a binary response variable (0 = incorrect belief, 1 = correct belief). We included the predictors condition (Group, No Group, factor), age (4–6, continuous numeric), and their interaction as fixed effects. We followed the same analysis plan as for the number of boxes opened.

What children believed about the boxes depended on whether they belonged to a group (see Fig. 2B). The full model for belief was a significantly better fit than the null model ($\chi^2(3) = 13.24$, $p = 0.004$, $R^2 = 0.21$) and once again, we found a significant effect of condition ($\chi^2(1) = 8.53$, $p = 0.003$, $R^2 = 0.14$), such that whether children believed there were more elephants or lions in the boxes depended on whether they were in the Group Condition or No Group Condition. Only about a quarter of children in the Group Condition correctly believed there were more lions in the boxes, while more than half of children in the No Group Condition believed so. In other words, children who belonged to a group were half as likely to hold an accurate belief about the boxes. We did not find a significant interaction effect between condition and age ($\chi^2(1) = 3.03$, $p = 0.082$, $R^2 = 0.05$), though we did find a significant effect of age ($\chi^2(1) = 4.55$, $p = 0.033$, $R^2 = 0.08$), such that older children more often held the incorrect belief that there were more elephants in the boxes.

We also conducted an exploratory analysis to assess the relationship between children's beliefs and the number of boxes opened. To do so, we conducted a mediation analysis to investigate whether the effect of condition on belief was statistically mediated by the number of boxes opened (see Fig. 2C). Condition (Group, No Group, factor) was treated as the intervention variable, the number of boxes opened as the mediator, and the belief formed as the outcome variable. This analysis was carried out using the mediation package (version 4.5.1)[82]. This analysis revealed that the group manipulation effect on the outcome was significantly mediated through the number of boxes that children opened, as indicated by the significant average mediation effect (ACME = −0.119, 95% CI [−0.238, −0.011], $p = 0.030$). The total effect of the condition (including direct and indirect effect) on belief formed was significant (Estimate = −0.252, 95% CI [−0.460, −0.030], $p = 0.027$), while the direct effect (ADE, c') of condition on belief alone was not significant once the mediation variable was considered (ADE = −0.133, 95% CI [−0.328, 0.063], $p = 0.186$).

## Study 2: How children weigh evidence that supports their group's belief

Why do children open fewer boxes to arrive at a conclusion in the Group Condition? We hypothesized one primary driver is that children place more weight on evidence supporting their group's belief. To test this, in Study 2, we examine whether 4–6-year-olds are more convinced by group-supporting evidence. Just as in Study 1, children first either joined one of two groups (in the Group Condition) or joined neither group (in the No Group Condition). They were then introduced to the task, wherein they needed to figure out whether there were more elephants or lions in a set of ten boxes. Children were told the two groups held different beliefs about the boxes. Following this, children saw two out of the ten boxes (e.g., two elephants) which supported their group's belief in the Group Condition or group 1's belief in the No Group Condition. Then, children were asked our main dependent variable: how confident they were that there were more elephants (level of confidence was expressed on a nine-point temperature scale, see Fig. 3B). We hypothesized that children in the Group Condition would demonstrate stronger confidence in their belief following exposure to evidence relative to children in the No Group Condition. In other words, group membership would lead children to weigh evidence more heavily when it supported their group's belief.

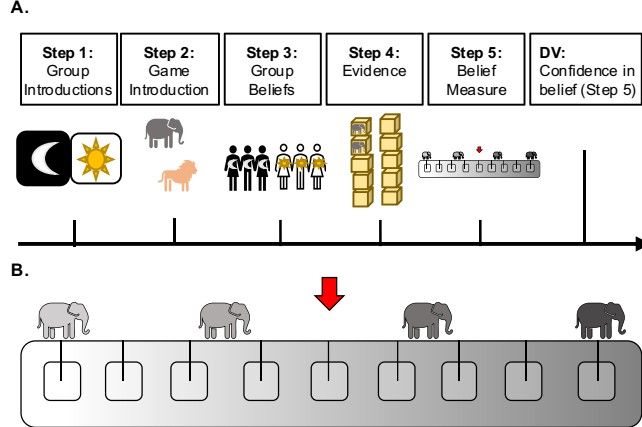

**Fig. 3 | Procedure and confidence scale used in Study 2. A** The five-step procedure used in both conditions: Step 1: Children either joined one of two groups or were simply introduced to them. Step 2: Children were tasked with figuring out whether there were more elephants or lions in a set of ten boxes. Step 3: Children were told about the two groups' contrasting beliefs about the boxes. Step 4: Children were shown two out of the ten boxes. Step 5: Children indicated how confident they were that there were more elephants on a nine-point temperature scale. **B** Temperature scale used to assess children's confidence in their beliefs in Step 5: Children moved a red marker onto one of nine squares. The far-left square indicated children were not sure there were more elephants, the middle-left squares indicated children were kind of sure, the middle-right squares indicated children were mostly sure, and the far-right square indicated children were really sure.

As pre-registered, to assess how confident children were following exposure to evidence, we fit a linear model predicting children's confidence (1–9, continuous numeric) by condition (Group, No Group, factor). Children in the Group Condition showed stronger confidence after seeing evidence than children in the No Group Condition (see Fig. 4A and B). The full model containing condition (Group, No Group) was a significantly better fit than the null model ($\chi^2(1) = 128.03$, $p < 0.001$, $R^2 = 0.11$). Children in the No Group Condition possessed an average confidence of 5.13 units (SD = 3.16), while children in the Group Condition possessed an average confidence of 7.16 units (SD = 2.62).

In an additional exploratory analysis, we built a linear model predicting confidence by condition (Group, No Group, factor), age (4–6, continuous numeric), and their interaction (like our full models in Study 1). This model was significantly different compared to a null model containing only the intercept ($\chi^2(3) = 133.22$, $p < 0.001$). Neither the interaction effect between condition and age ($\chi^2(3) = 1.15$, $p = 0.709$), nor the main effect of age were significant ($\chi^2(1) = 4.03$, $p = 0.485$). However, the condition effect remained significant even when controlling for age ($\chi^2(1) = 125.73$, $p < 0.001$).

## Study 3: How children weigh evidence that opposes their group's belief

In Study 3, we investigated whether 4–6-year-olds are less convinced by evidence that opposes their group belief. Just as in Study 1 and 2, children first either joined one of two groups (in the Group Condition) or joined neither group (in the No Group Condition), and were then introduced to the task, wherein they needed to figure out whether there were more elephants or lions in a set of boxes. Children were told the two groups held different beliefs about the boxes. This time, immediately following the group belief stage, children were asked about their own belief and the confidence they had in that belief (confidence was expressed on a nine-point temperature scale, see Fig. 5B). Following this, children saw two pieces of evidence that opposed their initial belief. Then, children were asked again what they believed and how confident they were in that belief. Our

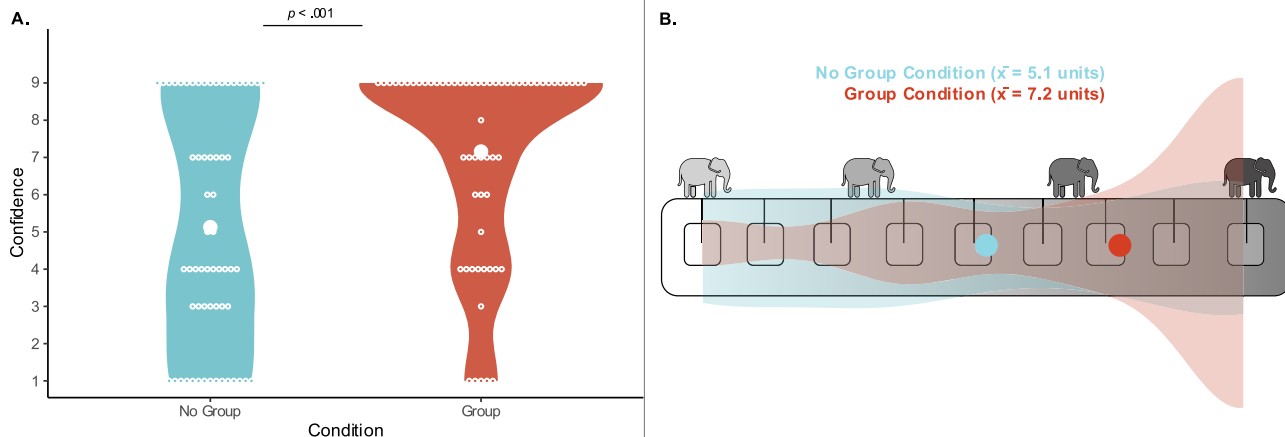

**Fig. 4 | Group influence on children's evaluation of group-supporting evidence in Study 2. A** Children's confidence in their belief after seeing evidence by condition: The distribution of responses for children in the No Group Condition is in blue, the distribution of responses for children in the Group Condition is in red. Individual data points (open circles) along with the means (closed circles) by condition are in white. Statistical test: model comparison testing the effect of condition as described in the main text (two-sided test; $\chi^2(1) = 128.03$, $p < 0.001$, $R^2 = 0.11$; $n = 124$ biologically independent replicates). To account for multiple comparisons across panels, we first conducted a full-null model comparison before testing individual effects. **B** Children's mean confidence ratings: Blue and red circles represent confidence ratings by condition, respectively. Overlaid are children's distribution of responses in (**A**). The far-left square indicated children were not sure, the middle-left squares indicated children were kind of sure, the middle-right squares indicated children were mostly sure, and the far-right square indicated children were really sure.

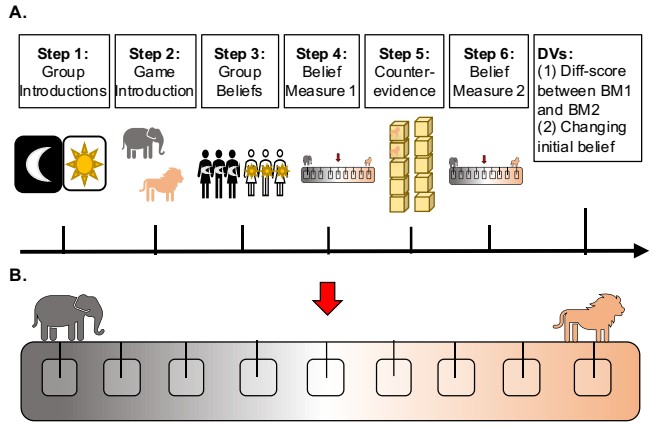

**Fig. 5 | Procedure and confidence scale used in Study 3. A** The six-step procedure used in both conditions: Step 1: Children either joined one of two groups or were simply introduced to them. Step 2: Children were tasked with figuring out whether there were more elephants or lions in a set of ten boxes. Step 3: Children were told about the two groups' contrasting beliefs about the boxes. Step 4: Children indicated how confident they were in their belief on a nine-point temperature scale. Step 5: Children were shown two out of the ten boxes that opposed their belief. Step 6: Children indicated how confident they were following exposure to counterevidence. Note BM1 and BM2 stand for Belief Measure 1 and Belief Measure 2, respectively. **B** Temperature scale used to assess children's confidence in their beliefs in Steps 4 and 6: Children moved a red marker onto one of nine squares. The far-left and far-right squares indicated children were really sure there were more elephants or lions, respectively. The middle-left and middle-right squares indicated children were kind of sure there were more elephants or lions, respectively. The middle square indicated that children did not know.

main dependent variable was the difference between children's initial rating of confidence and their second rating of confidence. We hypothesized that children in the No Group Condition would show a stronger reduction in confidence in their initial belief following exposure to counterevidence relative to children in the Group Condition. In other words, group membership would lead children to discount evidence more heavily when it went against their group's belief.

First, we looked at children's initial confidence rating. We assessed how confident children were in what they believed before seeing counterevidence. We built a linear model predicting children's initial confidence (initial distance from the midpoint, 0–4) by condition (Group, No Group). There was no statistically significant difference between the full model containing condition and the null model ($\chi^2(1) = 0.82$, $p = 0.618$, $R^2 < 0.01$). Children in the No Group Condition were on average 2.6 units confident in their beliefs (SD = 1.9), and children in the Group Condition were on average 2.3 units confident (SD = 1.8).

To assess whether group membership affected children's evaluation of counterevidence, we fit a linear model predicting reduction in confidence (calculated as a difference score between their Belief Measure 1 and 2 ratings) by condition. Children in the No Group Condition showed a stronger reduction in confidence after seeing counterevidence than children in the Group Condition (see Fig. 6A, B). The full model containing condition (Group, No Group, factor) was a significantly better fit than the null model ($\chi^2(1) = 60.00$, $p = 0.019$, $R^2 = 0.09$). Children in the No Group Condition moved an average of 4.2 units (SD = 3.9) towards the belief supported by the counterevidence, while children in the Group Condition moved an average of 2.2 units (SD = 2.6). This indicates that while children were not initially more convinced by their group's belief (see paragraph above), they were less convinced by evidence that contradicted the group belief.

In an exploratory analysis, we also fit a linear model predicting reduction-in-confidence by condition (Group, No Group, factor), age (4–6, continuous numeric), and their interaction (like our full models in Study 1 and the exploratory analysis in Study 2). The full model did not significantly improve model fit over the null model ($\chi^2(3) = 76.44$, $p = 0.072$, $R^2 = 0.11$) and neither the interaction effect between age and condition ($\chi^2(1) = 15.48$, $p = 0.221$, $R^2 = 0.02$), nor the main effect of age were significant ($\chi^2(1) = 0.97$, $p = 0.761$, $R^2 < 0.01$). However, the condition effect remained significant when controlling for age ($\chi^2(1) = 60.89$, $p = 0.018$, $R^2 = 0.09$).

Next, in an additional exploratory analysis, we measured whether children differed across conditions in how often they revised their initial belief. We fit a generalized linear model predicting belief revision, a binary response variable (0 = no belief revision, and 1 = belief revision) predicted by condition (Group, No Group, factor) and age

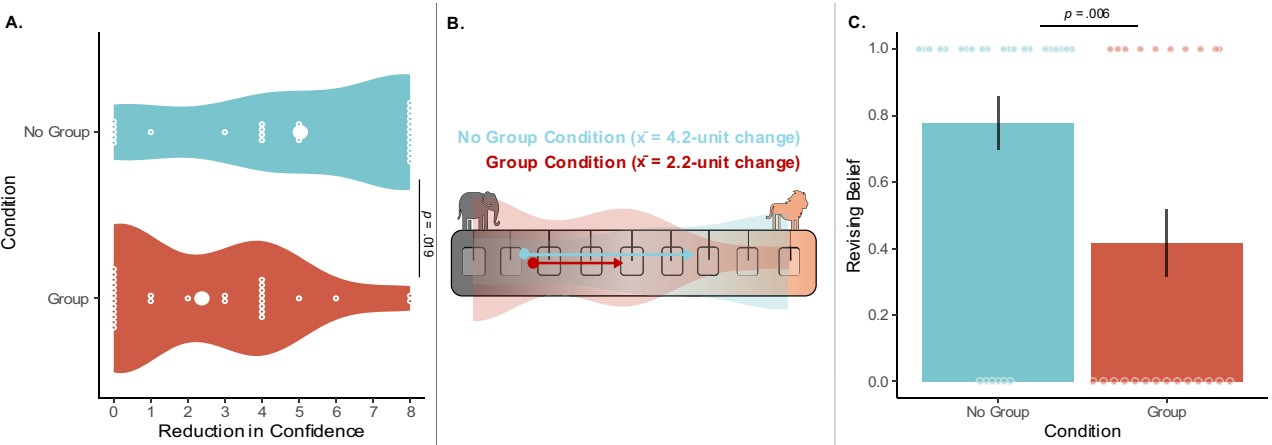

**Fig. 6 | Group influence on children's evaluation of group-opposing evidence in Study 3. A** Unit reduction in confidence in children's belief after seeing counterevidence by condition: The distribution of responses for children in the No Group Condition is in blue, the distribution of responses for children in the Group Condition is in red. Individual data points (open circles) along with the means (closed circles) by condition are in white. Note that three children who increased in confidence after seeing counterevidence were removed from this plot (but not from our analyses) for visual simplicity. Statistical test: model comparison testing the effect of condition as described in the main text (two-sided test; $\chi^2(1) = 60.00$, $p = 0.019$, $R^2 = 0.09$; $n = 60$ biologically independent replicates). To account for multiple comparisons across panels, we first conducted a full-null model comparison before testing individual effects. **B** Children's initial confidence ratings and mean unit change towards the belief supported by counterevidence: Blue and red circles represent children's initial ratings of confidence by condition; arrows represent means for the unit-difference between children's initial and secondary ratings of confidence by condition. Note that means include both when a child's initial belief was elephants or lions (represented on only the left side of the scale for visual simplicity). Overlaid are children's distribution of responses in (**A**). The far-left and far-right squares indicated children were really sure there were more elephants or lions, respectively. The middle-left and middle-right squares indicated children were kind of sure there were more elephants or lions, respectively. The middle square indicated children did not know. **C** Proportion of children revising their belief about the boxes after seeing two boxes of counterevidence: Data are presented as mean values +/- SEM, $n = 51$ biologically independent replicates. Statistical test: model comparison testing the effect of condition as described in the main text (two-sided test: $\chi^2(1) = 7.46$, $p = 0.006$, $R^2 = 0.18$; $n = 51$).

(4–6, continuous numeric). The full model was a significantly better fit than the null model ($\chi^2(2) = 1.66$, $p = 0.023$, $R^2 = 0.14$). Next, we compared the full model to respective reduced models without the predictor of interest. We found a significant effect of condition ($\chi^2(1) = 7.46$, $p = 0.006$, $R^2 = 0.18$), such that children in the No Group Condition more often revised their beliefs (see Fig. 6C). We did not find a statistically significant effect of age ($\chi^2(1) = 0.01$, $p = 0.910$, $R^2 < 0.01$).

## Discussion

Our results demonstrate that group membership influences young children's evidentiary standards, which in turn shapes how children form and revise their beliefs. In Studies 1 and 2, 4–6-year-old children who belonged to a group lowered their standards of evidence to form group beliefs when they encountered evidence that supported what their group believed. In Study 3, 4–6-year-old children who belonged to a group raised their standards of evidence when they encountered evidence that contradicted what their group believed. Therefore, although children are often compared to scientists who are motivated to seek out and test the available evidence in order to hold correct beliefs about the world[13–19], our results demonstrate that much like scientists in the real world, children are also social beings. When children belong to a social group, their standards of evidence shift, resulting in beliefs that align with their group. This indicates that the motivation to hold group beliefs emerges early in development, shaping the beliefs that even preschoolers adopt. Below, we provide a more detailed discussion of our findings and propose directions for future research.

Study 1 and 2 revealed a minimal group effect on children's belief formation practices. In Study 1, children were exposed to a set of evidence that initially seemed to support their ingroup's belief (for children who belonged to a group), but, overall, supported the outgroup's belief. Children who belonged to a group inspected fewer pieces of evidence to reach a conclusion compared to those who were exposed to the same information about the two groups' beliefs but did not

belong to either group. Importantly, these biased epistemic practices significantly impacted the accuracy of children's conclusions. A mediation analysis revealed that group membership was tied to how much evidence children sought out, which in turn influenced their belief. Consequently, children in the Group Condition were about half as likely to arrive at the correct answer relative to children in the No Group Condition. This suggests that group membership may not have influenced children's beliefs directly but did so through their evaluation of evidence—decreasing the amount of group-supporting evidence children needed to acquire to adopt a group belief.

Study 2 revealed why children inspected fewer pieces of evidence in Study 1: because they were more convinced by evidence that supported their group's belief. In Study 2, all children were shown two pieces of evidence. Children were more confident after seeing this evidence if it supported a group belief. Therefore, Studies 1 and 2 indicate that children are more readily convinced by evidence supporting their group's belief, relative to evidence supporting a non-affiliated group's belief, which leads them to seek out fewer pieces of evidence to arrive at a conclusion. These results suggest that children place more weight on evidence when it supports an ingroup belief.

While Study 1 and 2 focused on belief formation, Study 3 revealed a minimal group effect on children's belief revision practices. After hearing both groups' beliefs, children committed to a belief themselves and rated how confident they were. There was no significant difference in children's initial confidence in their beliefs between the two conditions. Then, all children were shown two pieces of counterevidence. How children responded to the counterevidence varied as a function of group membership: children maintained a higher level of confidence in their initial belief if it was a group belief. In other words, children were less swayed by evidence when it misaligned with their group's belief. This led children in the Group Condition to revise their initial beliefs (e.g., switching from believing that the boxes contain more lions to believing that the boxes contain more elephants) about half as often relative to children in the No Group Condition. These

results suggest that children place less weight on evidence when it opposes an ingroup belief.

Taken together, the results of the Group Condition and the No Group Condition across Studies 1, 2, and 3 suggest that children are both motivated to hold accurate beliefs, and to hold group beliefs. When children did not belong to a group, they demonstrated rational epistemic practices and tended to arrive at accurate conclusions (as in Study 1). Indeed, children's strong epistemic motivations to get at the truth are perhaps best demonstrated by the fact that in Study 1, over a third of the children in the No Group Condition opened all ten boxes, and thus waited for more than 4 minutes, sitting idly, just to evaluate all of the existing evidence. However, when children belonged to a group in the Group Condition, their epistemic practices markedly changed. They inspected significantly fewer pieces of evidence to arrive at a conclusion if the evidence supported their group's belief, were more convinced by group-supporting evidence, and held onto group beliefs more strongly in the face of counterevidence.

These findings demonstrate that adjusting one's epistemic practices in group contexts has deep developmental roots. Already at four years of age children are biased towards adopting group beliefs, which markedly impacts how they evaluate evidence. Previous research has demonstrated that children rationally form beliefs and are motivated to test the available evidence[27]. Yet, we find that once preschool-age children become part of a group and become aware of what their group believes, they become biased in how they form beliefs and how they test evidence, even when their group is not physically present. From an early age, a core motivation behind belief formation appears to be to align our beliefs with our groups. This likely has important consequences for the types of information children trust[46,83], how they perceive certain events[80], and what they come to think is of value and importance[46] as they age. Future research should address these open questions.

On our interpretation, the current results are explainable in terms of children's biased evaluation of the evidence in the Group Condition. One might alternatively suggest that children simply claimed to hold the group belief without actually believing it, or held the group belief irrespective of the evidence. However, these alternatives do not fit the observed pattern of results well. The overwhelming majority of children in Study 1 denied that they knew whether there were more elephants or lions in the boxes and requested to wait 30 s to open the next box several times. On average, children in the Group Condition opened roughly four boxes and 84% of children opened at least two boxes. If children in the Group Condition had simply changed their publicly stated judgment, or formed a belief irrespective of the evidence, one may not expect them to be so patient and curious about the content of the boxes. Additionally, a mediation analysis indicated that the influence of condition on children's beliefs was significantly mediated by the number of boxes children opened, suggesting their belief depended on the evidence they opened, which was in turn shaped by whether they were part of a group. An even stronger reason to think that children in the Group Condition did not simply say something without believing it, or formed a belief irrespective of the evidence, comes from the results of Study 3, where, having heard both groups' beliefs, children's initial ratings of confidence were identical across the two conditions. This suggests that group membership did not simply lead children to adopt group beliefs regardless of the evidence; rather, group membership changed children's evaluation of the evidence, which in turn led them to form group beliefs.

Preschoolers possess a bias in how they evaluate evidence to form and maintain the same beliefs as their groups. At the same time, the effect of group membership did not significantly vary by age. One potential reason for the lack of an age effect is that we based our power analyses on expected main effects, and so we were potentially not properly powered to find an interaction effect. Another possible reason is that our age range was too narrow to capture a developmental

trajectory. For instance, in ref. 75, 7–9-year-olds were more disapproving of a group member who held a different belief than the member's group compared to 4–6-year-olds. It's possible that 7–9-year-olds may also be more motivated to align their beliefs with their groups relative to younger children. As children grow older, social identity and group belonging may become increasingly central to reasoning and decision-making, which could lead older children to more strongly prioritize group information. At the same time, children's reasoning abilities mature with age, potentially enabling more sophisticated assessments of evidence[84,85]. Future work should examine how the bias to hold group beliefs changes over time across a wider range of development and how this influences children's standards of evidence.

Nonetheless, in Study 1, irrespective of whether children joined a group, we found that older children were more likely to form the incorrect belief (e.g., that there were more elephants in the boxes). One low-level possibility is that older children found the game less entertaining than younger children. Another possibility is that they were less curious about the contents of the boxes. Although we did not find a statistically significant effect of age on children's evidence seeking ($p = 0.08$), prior research on children's epistemic practices has shown that older children and adolescents tend to exhibit reduced exploratory tendencies when forming beliefs[86–88], however, findings on age-related changes are mixed[32]. This work raises the possibility that, as children grow up, they may become less concerned with forming the most accurate beliefs, and instead, increasingly generalize based on the evidence they have already seen.

In the current studies, in light of information about group beliefs, children adjusted their standards of evidence. Children did not, however, completely compromise their evaluation of evidence to blindly hold group beliefs. As noted above, children in Study 1 often needed to see multiple pieces of evidence before committing to their group's belief (only 16% of children in the Group Condition opened only the first box). Additionally, in Study 3, children who received counterevidence that opposed their group belief adjusted their confidence in light of this evidence, just not to the same degree as children who didn't belong to a group. This indicates that, even in group contexts, children are still motivated to test evidence and, to a certain degree, appropriately update their beliefs in response to it. Yet, belonging to a group introduces an additional concern: to align their beliefs with what their group believes.

Why do children possess such a deeply rooted bias to hold group beliefs? At first pass, this bias may appear irrational as adopting group beliefs can sometimes result in holding inaccurate beliefs (as seen in Study 1). However, as noted in the Introduction, there may be at least two rational motivations for children to adopt group beliefs and bias their standards of evidence. First, children may do so because they are epistemically motivated and think their groups are reliable, accurate sources of information. This intuition may be rational as our social groups often possess correct information we do not have[44,89]. Therefore, children may preferentially adopt group beliefs as an epistemic shortcut: they might simply believe that their groups have correct beliefs. This would suggest the main aim of children's beliefs is to accurately map onto the true state of the world[2–5] and they employ various heuristics to this end.

Children may also adopt group beliefs because they are instrumentally motivated to hold useful beliefs. Holding a correct belief may not always be in one's self-interest. Indeed, holding incorrect beliefs can often provide many benefits that outweigh the costs[6,90,91]. For instance, beliefs can serve as signals of group membership and help us fit in, gain a sense of belongingness, and avoid ostracism[1]. Thus, children may adopt group beliefs because doing so can potentially lead to social benefits. Future research should tease apart the epistemic and instrumental motivations behind why children form group beliefs (and beliefs more generally), as well as how children prioritize forming

beliefs for their practical versus epistemic benefits across various contexts. One method of isolating children's instrumental motivations would be to manipulate whether children expect to interact with their group members, while holding any reason to trust them constant across conditions. We hypothesize that children who expect to interact with their group would have a heightened motivation to affiliate with them, which would bias their evaluation of evidence more relative to children who do not expect to interact with their group.

Finally, the current results may have important implications for interventions aimed at reducing the spread of misinformation and the harmful effects of group biases, as well as fostering intellectual humility and evidence-based reasoning skills[11,12,92–96]. Although not directly tested here, our findings suggest it could be helpful to implement critical thinking, information literacy, and psychological inoculation interventions in preschool-aged children, before partisan biases mature and become more entrenched in their identities and everyday belief formation habits[97]. For instance, intergroup contact in adolescence is associated with more positive intergroup attitudes over time, whereas similar effects are not observed in older cohorts[98]. Developing an early awareness of group biases may help children better evaluate when trusting group members is beneficial or harmful.

In summary, the current studies document an early-emerging tendency in children to adopt and retain group beliefs. From a young age, children adjust their standards of evidence to align their beliefs with their groups. This biased evaluation of evidence in turn leads children to hold group beliefs, even when presented with counterevidence.

## Methods
The design, procedure, and analyses for Studies 1, 2, and 3 were pre-registered at AsPredicted.org (Study 1: https://aspredicted.org/k48x-tmyf.pdf, 20th October 2022, Study 2: https://aspredicted.org/j7fc-b4ym.pdf, 23rd September 2024, Study 3: https://aspredicted.org/n5j3-jv4d.pdf, 30th June 2023). All analyses and accompanying predictions reported in the main text were pre-registered, unless otherwise specified. Data collection for Studies 1 and 3 began shortly before pre-registration (data from 18 of 78 children in Study 1 and 4 of 60 children in Study 3 had been collected at the time of pre-registration) as we were awaiting final confirmation of the pre-registrations from our research team and had limited availability of testing sessions during this period.

### Study 1
**Participants**. In Study 1, we tested $N = 78$ 4–6-year-old U.S. American children ($M = 5.53$ years, SD = 0.82, 44 girls). Participants' gender was reported by their parents. In all studies, participants' gender was not considered in the study design. No significant gender effects were found in the current analyses (see Effect of Gender under Supplementary Methods in the Supplementary Information). Participants were identified by their parents as being White (40%), Asian (27%), Hispanic or Latino (9%), multiple races (9%), and other/unknown (15%). Children were recruited from around the San Francisco Bay Area. As pre-registered, we aimed to include 80 participants in our final sample following a power analysis, which suggested we would have an average power of $1 - \beta > 0.82$ to detect a significant effect of condition. To reach our sample size, we tested 91 children. We excluded 13 children from our final analyses: 12 due to experimenter error and one due to parental interference, following our pre-registered criteria. Therefore, our final sample size was 78 children. In all studies, children received a sticker for compensation at the end of the experiment.

### Design and procedure
In Study 1, children in both conditions followed the same four-step procedure (see Fig. 1A). Children were first randomly assigned to either the Group Condition (in which they joined one of two groups) or the No Group Condition (in which they joined no group; between-subjects design). Experimenters were not blinded to condition allocation or outcome assessment. In the Group Condition, children could choose to join either the Sun or Moon group. When children joined a group, children were informed that they were now part of that group, and were given a group hat and sticker to wear for the duration of the experiment. In the No Group Condition, children were simply told that there were two groups. All children then saw pictures of three group members each (two females and one male in each group). To measure whether our group manipulation was successful, children were then asked how much they liked a group member from each group (on a 5-point scale from really dislike to really like) and were asked to divide five stickers between the group members (gender-matched; following Dunham et al.[50]). These exploratory results revealed consistent trends toward ingroup preference across studies, aligning with prior work[51] (please see Preference Measures under Supplementary Methods in the Supplementary Information).

Next, children were then introduced to the main task of the experiment, which was to figure out whether there were more toy elephants or more toy lions in a set of ten boxes on a table. The toy animals were ~1.5 × 1.5 in. in size, and the wooden boxes were approximately 3 × 3 in., stacked in two columns of five.

Children were then told that both groups had played this game earlier. Children watched pre-recorded videos from both groups in which the group members said, in unison, whether they thought there were more elephants or lions in the boxes (e.g., "As the Moon team, we believe there are more elephants in the boxes"). To simplify the design, the ingroup (for children in the Group Condition) or group 1 (for children in the No Group Condition) always believed there were more elephants in the boxes, and the outgroup or group 2 always believed there were more lions in the boxes.

Lastly, children were given the opportunity to evaluate evidence for themselves by opening the boxes one at a time. In between opening each box, however, children had to wait 30 s, measured with a small hourglass timer. All children started by opening the first box, which they did not have to wait to open. After each subsequent box, children were asked: "Do you know whether there are more elephants or lions in the boxes, or do you want to open another box?" If children said they knew, they were asked whether they believed there were more elephants or more lions in the boxes (if they did not already state elephants or lions). If children said they wanted to open another box, they waited 30 s and could then open the next box. This process continued until they either said they knew or they opened all 10 boxes. Importantly, the ten boxes were always arranged in the same order, such that the first five boxes children could open seemed to suggest that there were more elephants in the boxes (four elephants to one lion; supporting the ingroup belief). However, in total, there were more lions (four elephants to six lions; confirming the outgroup belief; see Fig. 1B).

Our two main dependent variables were: (i) how many boxes children opened and (ii) whether children incorrectly believed there were more elephants or correctly believed there were more lions in the boxes. We predicted that children in the Group Condition would open fewer boxes than children in the No Group Condition. We also predicted that children in the Group Condition would more often incorrectly believe there were more elephants in the boxes (which is what their group believed). Finally, we predicted this tendency might be stronger for older children than younger children.

### Coding
Coding of how many boxes children opened (1–10, numeric) and children's belief (0 = incorrect belief, 1 = correct belief) was completed by the first author during the sessions with children and later from video recordings. An independent research assistant who was blind to the design and hypotheses coded 25% of trials. The reliability between raters was perfect (ICC = 1.00).

## Deviations from preregistration

In Study 1, we pre-registered building a generalized linear mixed model[97] to predict the number of boxes children opened by condition. In this model, opening each box was treated as a binary response variable (yes/no), with each participant having the possibility of opening 10 total boxes. We included the predictors condition, age, and their interaction as fixed effects, and a random intercept for individual identity. For consistency with the other analyses in the manuscript, we report results from a simplified linear model in the main text. Importantly, the conclusions do not differ between the two approaches (for more details about the initially planned analysis, please see Study 1 under Supplementary Methods in our Supplementary Information).

## Study 2

**Participants.** We tested $N = 124$ 4–6-year-old U.S. American children ($M = 5.56$ years, SD = 0.91, 51 girls). Participants' gender was reported by their parents. Participants were identified by their parents as being White (29.8%), Asian (23.4%), Hispanic or Latino (6.5%), African or African American (4.0%), multiple races (16.1%), and other/unknown (20.2%). Children were recruited from around the San Francisco Bay Area. We aimed to include 124 participants in our final sample, determined by a power analysis based on pilot data from 27 children. In the pilot, children were randomly assigned to either the Group Condition or No Group Condition, and we measured their confidence ratings after seeing evidence. The Group Condition had a mean confidence rating of 7.31 (SD = 2.56), while the No Group Condition had a mean of 5.85 (SD = 3.29). We fit a linear model predicting confidence from condition, which yielded $R^2 = 0.11$, corresponding to Cohen's $f^2 = 0.13$. We used this effect size in a power analysis via pwr.f2.test() from the pwr package (version 1.3.0), which indicated that 124 participants would be sufficient for 80% power at $\alpha = 0.05$. To reach our sample size, we tested 139 children. In line with our pre-registered exclusion criteria, we excluded 15 children from our final analyses: two children due to experimenter error, seven children for failure to understand the confidence scale, one child who was not able to follow along with the procedure, and five children because we had already reached our predetermined sample size. Therefore, our final sample size was 124 children.

## Design and procedure

Children in both conditions followed a similar procedure as in Study 1 (see Fig. 3A): they were first randomly assigned to either the Group Condition or the No Group Condition, in a between-subjects design. Children were then introduced to the main task of the experiment to figure out whether there were more toy elephants or more toy lions in a set of ten boxes on a table.

Before hearing both groups' beliefs about the boxes, children were introduced to a nine-point temperature scale used to assess their confidence in their belief about the content of the boxes (adapted from previous research measuring children's confidence[99,100]). As seen in Fig. 3B, the far-left square indicated children were not sure there were more elephants, the middle-left squares indicated children were kind of sure, the middle-right squares indicated children were mostly sure, and the far-right square indicated children were really sure. Children were required to pass four check comprehension questions about the scale before continuing ("where would you place your marker if you were really sure there were more elephants?", "where would you place your marker if you were mostly sure there were more elephants?", "where would you place your marker if you were kind of sure there were more elephants?", and "where would you place your marker if you weren't sure there were more elephants?"). If children did not answer or answered incorrectly, instructions were repeated up to four times. If children still did not answer correctly, they were excluded ($N = 7$).

After hearing both groups' beliefs (where the ingroup in the Group Condition or group 1 in the No Group Condition believed there were more elephants), children were shown two of the ten boxes. They saw that both boxes contained an elephant. Following the opening of these two boxes, children were asked our main dependent variable, which was to indicate how confident they were that there were more elephants on the nine-point temperature scale. We predicted that children in the Group Condition would show stronger confidence in their belief following exposure to evidence, as compared to children in the No Group Condition.

## Coding

Coding of children's confidence rating (1–9, representing each square on the scale) was completed by the second author during the sessions with children and later from videotapes. An additional research assistant who was blind to the design and hypotheses independently coded 25% of trials. The reliability between raters was perfect (ICC = 1.00).

## Study 3

**Participants.** We tested $N = 60$ 4–6-year-old U.S. American children (M = 5.52 years, SD = 0.87, 26 girls). Participants' gender was reported by their parents. Participants were identified by their parents as being White (33.3%), Asian (18.3%), Hispanic or Latino (20%), African or African American (1.7%), multiple races (18.3%), and other/unknown (8%). Children were recruited from around the San Francisco Bay Area. We aimed to include 60 participants in our final sample, determined by a power analysis based on pilot data from 27 children. In the pilot, children were randomly assigned to either the Group Condition or No Group Condition, and we measured their difference scores after seeing counterevidence. The Group Condition (M = 3.57, SD = 3.76) showed less confidence in the counterevidence than those in the No Group Condition (M = 6.15, SD = 2.73). This yielded a Cohen's $d$ of 0.79, which we used for our power analysis via pwr.t.test() from the pwr package (version 1.3.0). This analysis indicated that a total sample size of 27 participants per group would be sufficient for 80% power at $\alpha = 0.05$, which we conservatively rounded to 30 participants per group in our pre-registration. To reach our sample size, we tested 75 children. In line with our pre-registered exclusion criteria, we excluded 15 children from our final analyses: five children due to experimenter error, five children for failure to understand the confidence scale, four children in the Group Condition who did not initially believe their group (as we were interested in how children evaluate counterevidence that went against a held ingroup belief), and one last child because we had already reached our predetermined sample size. Therefore, our final sample size was 60 children.

## Design and Procedure

Children in both conditions followed a similar procedure as in Study 2 (see Fig. 5A). Children were randomly assigned to either the Group Condition or the No Group Condition, in a between-subjects design. They were then introduced to the main task of the experiment, which was to determine whether there were more toy elephants or more toy lions in a set of ten boxes on a table.

Before hearing both groups' beliefs about the boxes, children were introduced to a nine-point temperature scale used to assess their confidence in their belief about the content of the boxes (adapted from previous research measuring children's confidence[99,100]). As seen in Fig. 5B, the far left and right squares of the scale indicated children were really sure there were more elephants or lions, respectively. The middle left and right squares indicated children were kind of sure there were more elephants or lions, respectively. The middle square indicated children did not know. Children were required to pass three check comprehension questions about the scale before continuing ("where would you place your marker if you really thought there were

more elephants?", "where would you place your marker if you really thought there were more lions?", and "where would you place your marker if you didn't know?"). If children did not answer or answered incorrectly, instructions were repeated up to four times. If children still did not answer correctly, they were excluded ($N = 5$).

After hearing both groups' beliefs (counterbalanced with regard to what the groups believed), children were asked what they themselves believed and to indicate how confident they were on the scale. Next, children were shown two boxes that opposed their belief. For example, if a child initially thought there were more elephants, in this step, they would be shown two boxes that had a lion in them (and vice versa). Following the opening of these two boxes, children were then asked again what they believed and to indicate how confident they were. Our main dependent variable was the difference between children's initial rating of confidence and their second rating of confidence. We predicted that children in the No Group Condition would show a stronger reduction in confidence in their initial belief following exposure to counterevidence, as compared to children in the Group Condition. An exploratory measure also examined whether children in the No Group Condition more often revised their initial belief (i.e., a child initially indicating a belief that there were more elephants, and changing to believing there were more lions, and vice versa).

### Coding
Coding of children's initial and secondary ratings of confidence (1–9, representing each square on the scale, numeric) was completed by the first author and a research assistant during the sessions with children and later from videotapes. An additional research assistant who was blind to the design and hypotheses independently coded 25% of trials. The reliability between raters was excellent (ICC = 0.99). For the analysis of children's initial confidence ratings below, we converted their initial ratings into distance from the midpoint scores (0–4), such that children who initially believed there were more elephants or lions in the boxes were on the same scale.

To analyze how children's confidence ratings changed following counterevidence, we calculated the difference between their two ratings. A change towards the belief supported by counterevidence was coded as a positive unit difference, a change towards one's initial belief was coded as a negative unit difference (note that all children except for three moved towards the belief supported by counterevidence in their secondary rating).

Lastly, to assess whether children revised their beliefs (e.g., switching from believing there were more elephants to more lions, or vice versa), we coded a change across the midpoint as 1 (belief revision) and no change across the midpoint as 0 (no revision). For interpretability, children whose second rating landed at the midpoint were not included in this analysis ($N = 9$). It is worth noting the conclusions remain unchanged if these children are instead coded as having revised their beliefs.

### Ethical Statement
Study protocols were approved by the Committee for the Protection of Human Subjects at the University of California, Berkeley (CPHS Protocol Number: 2019-10-12605). Informed written consent was obtained from all parents of children, and additional verbal assent was obtained from children who participated.

### Reporting summary
Further information on research design is available in the Nature Portfolio Reporting Summary linked to this article.

## Data availability
The anonymized raw data used in the analyses have been deposited in the Open Science Framework (OSF) project link at https://doi.org/10.17605/OSF.IO/QP6WC[101]. All figures and tables in the main text and Supplementary Information can be reproduced using the shared data and code.

## Code availability
The code used to analyze the data is available at the same repository as the data: https://doi.org/10.17605/OSF.IO/QP6WC[101]. All analyses were conducted in R (version 4.5.1) using the following packages: tidyverse (2.0.0), lme4 (1.1-37), car (3.1-3), pwr (1.3-0), rsq (2.7), partR2 (0.9.2), MuMIn (1.48.11), mediation (4.5.1), lmtest (0.9-40), dplyr (1.1.4), svglite (2.2.1), and ggbeeswarm (0.7.2).

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

## Acknowledgements

We thank all the families that participated in our studies. We also thank our research assistants, Maggie Debelak, Rachel Tran, Makena Umnas, Grace Yee, Sophia Calandrillo, Yash Gupta, Pavana Rajesh, and Alexandra Romero for their great support during data collection. J.E. acknowledges support from the Jacobs Foundation and from an NSF CAREER award (Award ID 2237075, JE). H.S. received funding from the European Union's Framework Program for Research and Innovation Horizon 2020 (2014–2020) under the Marie Sklowska-Curie Grant Agreement (Award ID 841021, HS).

## Author contributions

All authors contributed extensively to the work presented in this paper. J.C., A.C., D.A., H.S., and J.E. designed the experiments and prepared the

manuscript; J.C. and A.C. collected data; and J.C. and H.S. created the analytic models.

## Competing interests

The authors declare no competing interests.
