## [Transparent Peer Review file · Nature Communications]

Group Membership Biases Children's Evaluation of Evidence

Corresponding Author: Mr Joshua Confer

Version 0:

Reviewer comments:

Reviewer #1

(Remarks to the Author)

This is a provocative and interesting set of studies investigating the extent to which group membership induces confirmation bias in young children. In the first study, the authors find that 4-6-year-olds are less likely to test out evidence when they belong to a group with confirming evidence. In the second study, children reduced their beliefs less when they belonged to a group that provided evidence. Given the rise of political polarization and confirmation bias, this study is very much timely and represents an important step in our understanding of the early roots of how group membership and epistemic beliefs interact. Moreover, I found it clever and the manuscript to be well-written and easy to understand. All this is to say, I like these studies and manuscript a lot. Here are a few bigger recommendations for the paper:

(1) I'd like to see the authors discuss these findings in light of group conformity beliefs a little more. Namely, I thought about this paper a lot when I was reading these studies:

Haun, D. B., & Tomasello, M. (2011). Conformity to peer pressure in preschool children. *Child development*, 82(6), 1759-1767.

In the paper above, find that children don't really change their views very much - they change their public expression of their views. In the ms presented here, the experimenter doesn't have a group, but is presumably responsible for assigning children to the group. I'd like to hear whether children's responses are therefore shifts in belief, or shifts in what they believe is the proper expression.

To be clear, I don't think this makes the authors' findings any less interesting or important - simply that we need to better understand what they might or might not mean.

(2) Secondly, I am wondering if the authors could talk a little more about whether children are more quick to accept confirming evidence in the Group case, or is it that they are less likely to look for disconfirming evidence. Perhaps the distinction is subtle, but I think it's an important one: it's possible that upon initial confirmation of the group's claims, children are no longer motivated to seek out further evidence. In contrast, it's also possible that children are not necessarily accepting the group's claims, but are nervous to find disconfirming evidence of it. I don't think the data (or any possible follow-up study I can think of) can really tease apart these possibilities (and they are not exactly mutually exclusive). Still, I'd like to hear the authors' views on this, and if there are qualitative answers that were provided that shed light on this issue.

(3) I have a nitpicky point about the scale in Exp 2: it looks like a scale for how many more elephants vs. lions there are, rather than a confidence scale. I get what the authors were going for, but I wonder if children might have shared this confusion. If not, perhaps clarify what the training looked like.

(4) I'd also like the authors to consider some of the literature on testing counterintuitive claims - namely, the development of how they do so (older children are the ones who test counterintuitive claims, younger children tend to accept them), and integrate them into their discussion. Related papers below:

Ronfard, S., Chen, E. E., & Harris, P. L. (2021). Testing what you're told: Young children's empirical investigation of a surprising claim. *Journal of Cognition and Development*, 22(3), 426-447.

Ronfard, S., Chen, E. E., & Harris, P. L. (2018). The emergence of the empirical stance: Children's testing of counterintuitive

claims. Developmental Psychology.

Reviewer #2

(Remarks to the Author)

This paper tests a neat idea that even children as young as four years old will hold on to a belief for social reasons - to affiliate with their group.

The noteworthy results from experiment 1 that group affiliation restricts children's curiosity to find things out for themselves. This restricted curiosity causes them to form a belief which is accurate (rational) based on the evidence they chose to gather (limited), but inaccurate to the full set of evidence available (all the boxes on the table). The noteworthy results from experiment 2 are that children remained confident in their groups belief in the face of counterevidence, and were less likely to revise it, then when the belief belonged to a group which was not theirs (no group condition).

This paper is the first to show biased belief formation and revision based on group identity at age four. I think it on the surface it is both original and likely to be of wide interest. The paper is also well written and the conclusions are mostly (though see below for a minor quibble about experiment 1) supported by the evidence.

I have a few points that I believe could be addressed in a revision, perhaps with more data (or not, depending on how my concerns change the strengths of the conclusions):

1) I wish there were more age groups for comparison, and an adult comparison group, so that the authors could also explore potential developmental changes and mechanisms of change. For example, it seems to me that the initial belief is formed quickly because it exploits children's tendency to be credulous of in-group members (e.g. Kinzler, Corriveau et al) and conformist under pressure (Haun, Corriveau, etc). But older children (above age 6) may be less credulous initially than younger children, and more willing to trust themselves to look for evidence. Importantly, older children also have better probabilistic reasoning skills, and thus have a better grasp of the uncertainty involved about boxes that remain closed despite a seemingly deterministic pattern in the initial 3. Older children also are more patient, and can wait longer than 30 seconds for something they are interested in doing. All of this can potentially influence how far children will go to open boxes.

2) Relatedly, as I mentioned above, the simple description of the result was not that children "adjust their standards of evidence in order to hold the same beliefs as their group" but that children in the group condition explored fewer boxes (were less curious) than children in the no group condition. The two groups' testimony is also a form of evidence: For example, Bridgers and colleagues found that one general claim (like "there are more lions than elephants") is worth 3 or 4 pieces of observed "data." In this context, the number of boxes that constitutes the "standard of evidence" is unclear. This may be especially true if you are young, impatient, and not versed enough in probabilistic thinking such that you may expect an initially deterministic pattern (lion, lion lion...) to repeat.

3) The point above also holds for experiment 2 - it could be interpreted through this social learning lense as a finding that testimony from groups weights heavier than one's own observations. This could be rational for children to have evidential standards for testimony which weight it higher (e.g. Bridgers et al, also Sobel & Kushnir)

4) Finally, if the basic result of Experiment 1 is simply that group testimony restricts curiosity, it straightforwardly follows from the exploratory play literature (e.g. Bonawitz, Shafto, et al) - children explore less when adults present their knowledge confidently. Even though this is an extension of prior work, I agree that it is a very interesting finding in the context of groups. However, I'm not sure it is sufficient to warrant publication in this journal, at least not without a more complete picture of how much this result actually represents "political" reasoning as opposed to a social learning finding that follows from prior work.

Reviewer #3

(Remarks to the Author)

In this ms., the Authors present two studies with young children, suggesting that the children seek evidence and revise their beliefs in a biased manner, giving preferential treatment to beliefs held by members of a group they have been arbitrarily assigned to by contrast with the beliefs of a group they do not belong. This bias is not observed when the beliefs are held by two groups, neither of which include the child.

This ms. has very significant strengths. It tackles an important issue, it's well written, and the experiments are very clever.

This ms. also has some weaknesses, one quite glaring and the other minor.

The glaring weakness is that the Authors have completely overlooked several very relevant (and huge!) strands of literature. The Authors state that “Previous research on children’s belief formation practices has almost exclusively investigated how children form beliefs in individual contexts, in which participating children adopt beliefs on their own, with no social influences or other agents involved.” I really don’t understand how it’s possible to state that when there’s a massive literature on how children evaluate information communicated by others (the literature on trust in testimony, which is very much about belief formation in a social context). The Authors might say that, in that literature, the children are told something by someone, but then make the decision of whether to accept the belief on their own – but that’s essentially also what happens in the present experiment (the other children aren’t actually there, the child hasn’t interacted with them, etc.).

There’s also a literature on peer pressure in children that age, for instance: Haun, D. B., & Tomasello, M. (2011). Conformity to peer pressure in preschool children. *Child development*, 82(6), 1759-1767. There are dozens of papers in that literature.

The abstract of that paper reveals how relevant this strand of literature is to the present ms.: “Both adults and adolescents often conform their behavior and opinions to peer groups, even when they themselves know better. The current study investigated this phenomenon in 24 groups of 4 children between 4;2 and 4;9 years of age. Children often made their judgments conform to those of 3 peers, who had made obviously erroneous but unanimous public judgments right before them. A follow-up study with 18 groups of 4 children between 4;0 and 4;6 years of age revealed that children did not change their “real” judgment of the situation, but only their public expression of it. Preschool children are subject to peer pressure, indicating sensitivity to peers as a primary social reference group already during the preschool years.”

While I still believe the present ms. makes a contribution, the Authors have to situate their findings in the relevant literature so we can actually gauge its novelty, and they avoid overselling it.

The Haun and Tomasello paper also raises the question of how much the children would actually ground their actions in their stated, group-consistent beliefs (something the Authors might discuss, but don’t have to address with new data).

A more minor issue is that the Authors misinterpret the lack of evidence for an interaction between condition and age as evidence of absence (e.g. “Interestingly, 4-6-year-olds’ epistemic practices were equally influenced by group membership.”). They don’t have a large enough sample to confidently conclude from the lack of a significant interaction that there is no interaction. In addition, the p values of most of these interactions are consistent trends in the direction of an interaction, suggesting that it’s quite likely that there is an interaction. Maybe being able to look at the data (a graph in ESM?) might help get a better sense of what’s going on.

Other comments:

When the Authors report the effects of the manipulation check, they might consider providing the data, and not just the inferential statistics, especially since some of the effects appear to be quite small.

Would it be possible to run an (exploratory) analysis looking at whether condition has an effect, once you condition on the number of boxes opened in Experiment 1? I.e. could we know whether the effect of condition on the final belief is entirely due to the number of boxes opened? That seems relevant to understanding what the children are doing.

“four children in the Group Condition who did not initially believe their group”

If feels a bit like cheating to test whether children tend to believe their group after having removed the children who don’t believe their group. I’m not sure how that can be handled, but I don’t think the Authors can just say that they removed the children that show their hypothesis wrong...

“First, children may do so because they are epistemically motivated and think their groups are reliable, accurate sources of information. This intuition may be rational as our social groups often possess correct information we don’t have (Kane et al., 2005). Therefore, children may preferentially adopt group beliefs as an epistemic shortcut: they might simply believe that their groups have correct beliefs. This would suggest the main aim of children’s beliefs is to accurately map onto the true state of the world (e.g., Fodor, 2001; Millikan, 1984; McKay & Dennett, 2009; Cowie, 2014), and they employ various heuristics to this end.”

I don’t think that interpretation is consistent with the findings the Authors present right before: if it were epistemic, the children should just believe their group more in all the situations they have a chance to, which they don’t do (i.e. “The overwhelming majority of children in Study 1 denied that they knew whether there were more elephants or lions in the boxes and requested to wait 30 seconds to open the next box several times. On average, children in the Group Condition opened roughly four boxes and 82% of children opened at least two boxes. If children in the Group Condition had formed a strong belief right away, one may not expect children to be so patient and curious about the content of the boxes. An even stronger reason to believe that children in the Group Condition did not form an especially strong initial belief comes from the results of Study 2, where, having heard both groups’ beliefs, children’s initial ratings of confidence were identical across the two conditions. This suggests that group membership did not simply lead children to adopt group beliefs irrespective of the evidence; rather, group membership changed children’s evaluation of the evidence.”)

An alternative, compatible with the Authors’ second interpretation, is that most children try to avoid antagonizing members of their group by disagreeing with them: they don’t look for too much evidence that might show them wrong (in Experiment 1), and they don’t outright say their group is wrong (in Experiment 2). But they don’t put any special stock in their group’s belief otherwise (as shown by the evidence quote above). That’s also compatible with the fact that, in Experiment 1, the Authors had to make gathering information quite costly, presumably because otherwise the kids were checking all the boxes, even in the in-group condition.

Reviewer #4

(Remarks to the Author)

This manuscript examines the development of young children’s beliefs in social contexts, testing potential friction between accuracy motives (i.e., to form accurate beliefs about the world) and social motives (e.g., to fit in). Across two preregistered studies, 4- to 6-year-old children joined a minimal group (Group Condition) or did not (Control Condition) and then evaluated evidence. In Study 1, children played a game where they determined whether there were more toy elephants or more toy lions in a group of 10 boxes. After choosing a minimal ingroup (or not, in the Control Condition), children heard what each group believed (re: elephants vs. lions), having played the game earlier. Children then opened the boxes themselves to evaluate the evidence. All children opened the first box and then had to wait 30 seconds in between opening each subsequent box. The first five boxes suggested that the child’s ingroup was correct. The study revealed a minimal group effect on children’s belief formation processes: Children in the Group Condition opened fewer boxes than children in the Control Condition, suggesting that group membership lowered children’s standard of evidence. In Study 2, children were again assigned to a Group or Control Condition. Then, using Study 1’s game set-up, they learned about each group’s beliefs of whether there were more elephants or lions in 10 boxes. Children then reported what they believed (and how confident they were in that belief) and then were shown two pieces of evidence going against their belief. Their beliefs and confidence were measured again. Children in the Group (vs. Control) Condition were less convinced by evidence that opposed their ingroup’s beliefs (that is, their confidence in their beliefs did not change as much), suggesting that group membership affected children’s belief revision practices.

There is a lot to like about this paper. It addresses an important and timely issue, does so in an innovative but methodologically careful and sound way, and both the results and writing are crystal clear. That said, a careful review of the paper led us (this paper is co-reviewed with an early career researcher) to several key questions that need to be addressed:

1) Is this paper of sufficiently broad interest and importance to warrant publication in Nature Communications?

The topic is clearly relevant to a broad readership, and this work integrates several compelling literatures in developmental psychology and beyond. That said, more work could be done to motivate why the questions asked ought to be studied through a developmental lens. Or, to put this another way, what does the general readership learn from a developmental paper on this topic, beyond what is already known within psychological science more broadly? There are many ways the authors may address this. For example, in the introduction they could more clearly delineate why this work is important from a developmental perspective, e.g. by integrating more literature across social and political psychology as well as judgment and decision-making, by discussing how interventions aimed at addressing problematic biases might be particularly effective when targeting children or youth, and/or by talking more about the specific outcomes these kinds of group biases might have specifically among children (just to give several possibilities). The paper currently reads as an excellent study of broad interest to developmental psychologists specifically, but interest beyond that area of study is less clear.

2) Is the conceptual framing of the paper coherent and sound, and does it adequately motivate these particular studies?

The conceptual framework lays out two broad sets of motivations that might underlie belief formation: epistemic motives and motives termed both “instrumental” and “social” in different parts of the paper. This framing is compelling, but it seems somewhat oversimplified, and potentially incorrect. There are a few issues here. First, a stark contrast is drawn between rational, child-as-scientist motivations and (at least implicit) irrational social or political motivations. But this is a false dichotomy. For one, scientists are also social beings—like all humans—with biases and conflicting motives, and do not simply pursue the truth with a dogged single mindedness. Indeed, anyone who has spent time as an academic scientist knows all too well how political it can be. The “child-as-scientist” perspective is very common in developmental work, but it might be worth softening this dichotomy or even considering removing it altogether. At the very least, discussing some nuance within this framing would clarify the paper’s contribution.

Second, and more specifically for this work, it’s not clear that the model of two competing motives both feeding into belief formation independently, or the model of them as two sides of a balance scale (it’s not clear which of these metaphors is closest to what the authors want to argue for) is right. There are at least two ways the mechanism of non-epistemic factors coming into play might work – one having to do with competing motives, as the paper suggests, but the other having to do with weighting evidence differently depending on its source or its alignment with group beliefs. Put another way, there is a difference between children deciding to believe something they aren’t sure is accurate, or deciding to persist in a belief that is aligned with their group and ignoring counterevidence vs. children deciding that their group in fact has the accurate beliefs. The latter is an epistemic process that is colored by social group membership, rather than a social or instrumental motive “winning out” over an epistemic one. As it stands, the paper presents only one view of how this might work, and does not discuss any relevant alternatives.

3) Is the interpretation of the results sound, and might there be alternative explanations that ought to be tested or discussed?

As mentioned above, the results themselves are crystal clear. However, the extent to which they support the paper’s claims is less clear. The paper claims that children are at least in part driven by social over epistemic motives. But there is no direct evidence of this. The discussion above of different perspectives could lead one to predict that the mechanism might be motivational, or might be playing out within children’s epistemic reasoning—that is, children are motivated to be accurate, but their weighting of different evidence is influenced by group factors. That is a very different story, and one that is equally consistent with the data, at least on our reading. It would be helpful for the authors to lay out why they think the data support their interpretation and what data could clearly adjudicate between it and other explanations.

The interpretation of Study 2 is also a bit unclear. The group vs. no group effect in terms of the degree of belief revision is clear. The paper characterizes this as children in the group condition being resistant to counterevidence. But this may not be accurate. Children seem to move from being sure their group is right to completely unsure either way. That in itself is a belief revision that is consistent with the evidence they’ve seen, especially if they take their group-stated beliefs as equivalent to evidence (which they may, given that it is essentially testimony). Thus a more accurate interpretation might be something like children’s confidence in the ingroup-held belief is more deeply held than confidence in a belief stated by a group with whom they have no affiliation. But this is not the same as what the paper states. Softening the paper’s existing claims and discussing potential alternative interpretations will bring it more in-line with the data.

A few more specific comments are below. In sum, this paper has a lot to recommend it, but much work is needed to clarify its contribution and highlight its broad interest to the journal’s readership.

More specific comments:

1) p. 10, line 261: It seems that the ingroup liking manipulation check in Study 1 wasn’t significant but the resource allocation measure was. It’s unclear what to make of this, or how it bears on (or does not bear on) the results.

2) p. 12, line 309: Why would older children have more inaccurate beliefs?

3) p. 15, line 393: We think this should read “Step 5: Counterevidence” and a few lines down, “Step 6: Belief Measure 2” to be in-line with Figure 3.

4) Would we anticipate gender effects? More discussion would be helpful.

5) Did the authors have any parental data (e.g., parent political orientation)? If not, it might be worth discussing how parental factors might or might not be relevant here.

Reviewer #5

(Remarks to the Author)

Version 1:

Reviewer comments:

Reviewer #1

(Remarks to the Author)

I believe the authors have been very responsive to reviews and the additional study does help in clarifying the distinction between testimony restricting exploratory play and testimony changing the standards of evidence. The notable result is that group testimony changes one's standard of evidence for evaluating claims - I think we do not *yet* have a mechanism to explain why, but that is a tall order, and one reserved for future work!

Reviewer #3

(Remarks to the Author)

The Authors have done a good job of integrating the reviewers' feedback, including running a new and interesting study. I am now even more positive inclined towards the ms. However, I believe there are still significant issues of interpretation that ought to be addressed.

The results are sometimes framed in terms of demonstrating children put too much weight on the information from the in-group. However, and except in S1 (more on this below), there are no normative benchmarks. So it's equally possible that children are not putting enough weight on the no-group information. For instance, in S2, the children in the no-group condition are told the group's belief, then they see the evidence supporting that belief, and in spite of that they do not favor the group's belief at all (they are at the midpoint in terms of confidence). That suggests that they are very (overly?) skeptical of information from non-in-groups, instead of showing that they accept overly easily information from in-groups. So the conclusion they draw from S2 ("These results suggest that children overweight the evidentiary strength of evidence when it supports an ingroup belief.") seems to be completely backwards: the children underweight the evidence (and the opinion) of non-in-group members.

S1 is the study for which the Authors could make the strongest case for children overweighting in-group information by contrast with underweighting non-in-group information, since there's a normative benchmark (an accurate answer) that's accessible. However, to get there the set up has to be quite contrived, with a lot of preliminary evidence supporting the in-group belief, so that even a small bias in the direction of the in-group will yield inaccurate beliefs. It should be noted that the bias only leads to issues when the environment is somewhat deceptive, and that if that weren't the case, then it's not clear a difference in accuracy between the groups would have been obtained.

There might be something I'm missing, but I don't understand why the initial results from the no-group condition of S2 and S3 are so different. If I get it right, in S2, the children are told this group believes it's mostly elephants, then they are showed two elephants. With all this information, they still have no confidence at all that there are more elephants. By contrast, in S3, the children are only told of the group's belief, not shown any evidence, and they are nearly at ceiling in terms of believing that there are more elephants (for that matter, their belief seems to be higher even than their belief in the in group after the evidence has been shown in S2). What's going on? Why would showing two elephants lower the belief that there are elephants? Did it draw the children's attention to all the unopened boxes?

Finally, my main issue is with the framing of the conclusion. The Authors write for instance "When children belong to a social group, they adjust their standards of evidence in order to hold the same beliefs as their group." I don't think the evidence particularly supports this "in order to" interpretation. It could also be that children put more weight on information provided by their in-group (as suggested later on in the conclusion), or that they don't put enough weight on information provided by a non-in-group (see above). By analogy, imagine that the no-group condition were compared to an expert condition, in which someone who has, say, looked into the boxes, gives their opinion. If children accepted that expert opinion, it would obviously not show that they are motivated to hold the same beliefs as the expert—it would simply reveal that they are motivated to hold accurate beliefs (and that they trust the expert). If the children merely wanted to have the same belief as their in-group, they would have much easier ways of going about it (such as accepting the belief and not looking for any extra evidence). So I think the interpretation can't be that straightforward. This issue recurs throughout the discussion, and it really has to be fixed—the results just don't warrant that conclusion.

Minor comments

In the introduction, the Authors should be clearer about their theoretical framework: they go from belief formation in general to "how children evaluate evidence to form beliefs," which is a more specialized literature, back to belief formation more generally in the case of trust in testimony without explaining the differences between the objects of these literatures. So it's great that the Authors have added the literature that had been omitted the first time around, but it feels like it's been added maybe a bit thoughtlessly, without proper integration.

I think the Authors are overselling the potential of their findings for interventions that would have any type of effect in adulthood. To bolster their argument, they should mention studies that show such interventions are effective, otherwise I think it's too speculative (and the paper doesn't need that to be interesting).

"Why do children open less boxes" should that be "fewer"?

"epistemic weight" just weight (there are I believe a couple other instances of unnecessary 'epistemic')

The figure legend should indicate which study they refer to.

I'm not a stat expert, but it would be nice to have a sense of why the power analysis yielded a required sample size in S2 twice as large as in S3.

The main effect of age in S1 could also be explained by older children getting bored more easily by this game (waiting to look into boxes to find toy animals), so I don't think much stock should be put on this.

I couldn't find the pre-registration for the new study (there are only two pre-registrations in the link provided).

Reviewer #4

(Remarks to the Author)

Thank you for the opportunity to review this resubmission. Our review of the initial submission included several broader questions: (1) Is the paper of sufficiently broad interest and importance to warrant publication in this journal? (2) Is the conceptual framing of the paper coherent and sound, and does it adequately motivate these particular studies? and (3) Is the interpretation of the results sound, and might there be alternative explanations that ought to be tested or discussed? Having carefully re-read the revised manuscript and response letter, we believe that our initial reviews were appropriately addressed and think that this revision will make a strong contribution to the literature.

In addition to assessing the revision in light of our own reviews, we were asked to comment on whether Reviewer 2's reviews were appropriately addressed. In general, we believe that the revised manuscript addresses Reviewer 2's concerns, with two possible exceptions, which we discuss below.

First, Reviewer 2's Comment 2 asks about the possibility of testing additional age groups. This comment is addressed in part with the existing data, but it was unclear to us to what age group both the reviewer and response were referring when referencing "older children." It seems as though Reviewer 2 was talking about children over the age of 6, which—to our knowledge—the revision cannot currently speak to, even with the new data. Given this comment, it was surprising to us that the revision didn't include a slightly older age range in the new collected sample. Despite this, we do think that the revision is strong enough to warrant publication without new data examining additional age groups. The authors might consider justifying in a bit more detail why they focused on their young age range, and/or add to the General Discussion why (or why not) examining a broader age range could be an important future direction. The authors do this to some extent in the current General Discussion, but this could be expanded substantially.

Second, we weren't entirely sure how to interpret Reviewer 2's Comment 4. Our read of this comment seemed to be more about whether one can pull apart the influence of the groups' testimony from children's observation of evidence, both of which could be forms of evidence that shape children's responses. Based on the authors' response and our reading of the manuscript, it seems this may have stemmed from a misunderstanding. Specifically, R2's Comment 4 seems to suggest they understood that children in the no group condition did not receive testimony from a group, but as the authors' response clarifies, the manipulation was not about the source of the testimony, but rather whether or not children belonged to one of the groups—which in the no group condition, they did not. Thus our sense is that this comment has been addressed sufficiently.

We also have two new comments, largely in response to the added content:

1. The revision now includes an exploratory mediation in Study 1 (p. 15). We believe this is a cross-sectional mediation, but a lot of the language in Study 1 and the General Discussion surrounding this exploratory pattern sounds pretty causal—we would recommend softening the language, or more clearly articulating how causality can be inferred from this exploratory analysis.
2. Something about the combined presentation of Study 2's method across the figure and text made us initially a bit confused about when exactly the belief measure was administered. Upon a few re-reads, we assume it's after the evidence (two elephants) was provided. Also, in the figure, the belief measure (Step 5) and confidence in belief (DV) are the same thing, correct? Making the order of the measures clearer in text might allay potential confusion of future readers.

And lastly, a handful of minor comments:

3. At the end of Study 1's Introduction, the revision "predicted that this tendency would be stronger for older children than younger children (following the developmental trend in Roberts et al., 2021)" (p. 8). This "developmental trend" cited above wasn't really spelled out in the Introduction (unless we missed it), though it does appear in the General Discussion. Could this trend be more clearly articulated before the revision delves into the studies?

4. Below we detail what we think are several typos:

- a. Second-to-last line of Abstract: children' → children's
- b. Last line of p. 7 says that children were assigned to one of two groups. This is incorrect, right? Children chose one of two groups across studies?
- c. Last line of p. 17: Group Condition → No Group Condition (right?)

Thank you for the opportunity to review this revision.

Reviewer #5

(Remarks to the Author)

Version 2:

Reviewer comments:

Reviewer #3

(Remarks to the Author)

Thank you for your work on the revisions, I think it's considerably strengthened the ms.

Reviewer #4

(Remarks to the Author)

Thank you for the opportunity to review this revision. We believe that our reviews were appropriately addressed and think that this paper will make a strong contribution to the literature.

Regarding the editor's specific requests,

We don't have any major concerns about the discrepancy between the power analysis for Studies 2 and 3. That said, unless we missed it, there also isn't a lot of information in the manuscript to assess this discrepancy. For Study 2, the paper states, "We aimed to include 124 participants in our final sample, determined by a power analysis based on pilot data. The analysis suggested we would have an average power of $1-\beta > 0.80$ to detect a significant effect of condition." For Study 3, the paper states, "For Study 3, We aimed to include 60 participants in our final sample, determined by a power analysis based on pilot data. The analysis suggested we would have an average power of $1-\beta > 0.80$ to detect a significant effect of condition." Both studies seem to have primary DVs on comparable scales—perhaps Study 3 was powered with a larger effect in mind? The review response document states that "Our power analyses for both studies were based on pilot data, which yielded the respective sample sizes. Differences in the estimated variability or effect sizes between the pilot datasets likely contributed to the discrepancy between Study 2 and 3." This makes sense to us—the authors could consider adding more information about their power analyses (e.g., the effect sizes they were powering for) into the supplement or main text for further clarity.

2. We did not notice any issues regarding the information provided in the manuscript versus what was in the preregistrations. The comments about why data collection began before the preregistrations are quite common (in our experience)—especially when collecting data from children—and do not raise any red flags.

Reviewer #5

(Remarks to the Author)

RESPONSE TO REVIEWER 1:

- 1. This is a provocative and interesting set of studies investigating the extent to which group membership induces confirmation bias in young children... Given the rise of political polarization and confirmation bias, this study is very much timely and represents an important step in our understanding of the early roots of how group membership and epistemic beliefs interact. Moreover, I found it clever and the manuscript to be well-written and easy to understand. All this is to say, I like these studies and manuscript a lot.**

Response: We thank Reviewer 1 for this positive assessment of our work.

- 2. I'd like to see the authors discuss these findings in light of group conformity beliefs a little more. Namely, I thought about this paper a lot when I was reading these studies: Haun, D. B., & Tomasello, M. (2011). In the paper above, [they] find that children don't really change their views very much - they change their public expression of their views. In the ms presented here, the experimenter doesn't have a group, but is presumably responsible for assigning children to the group. I'd like to hear whether children's responses are therefore shifts in belief, or shifts in what they believe is the proper expression. To be clear, I don't think this makes the authors' findings any less interesting or important - simply that we need to better understand what they might or might not mean.**

Response: We thank Reviewer 1 for this comment, which has helped us clarify the theoretical contribution of our work. As suggested by Reviewer 1 (and Reviewer 3), we added a discussion of Haun & Tomasello (2011) in the introduction on page 5. As Reviewer 1 and 3 note, Haun & Tomasello (2011) indicate that children change their publicly stated views in response to social influences, but not their actual beliefs. One key difference is that unlike Haun & Tomasello (2011), in our studies, children belonged to a group, and we tested how this influenced their evaluation of evidence. Another key difference is that in the present manuscript, group membership changed children's actual beliefs. There are three main reasons to believe this:

1. In Study 1, children in the *Group Condition* opened roughly four boxes and 82% of children opened at least two boxes. If children in the *Group Condition* had simply changed their publicly stated judgment, we would not expect them to be so patient and curious about the content of the boxes. Additionally, in response to a suggestion by Reviewer 3 (please see our fifth response to R3 below), we ran a mediation analysis, which indicated that the influence of condition on children's beliefs was fully mediated by the number of boxes children opened. This suggests their belief was based on the evidence they opened, which was in turn influenced by whether they were part of a group or not.

2. In Study 3, after hearing both groups' beliefs, children's initial ratings of confidence were identical across conditions. Group membership only influenced how children evaluated the evidence itself. This suggests that belonging to a group did not simply lead children to adopt group beliefs blindly or simply state the group belief to the

experimenter without believing it; rather, group membership changed their evaluation of evidence, which in turn shaped what they believed.

3. The experimenter did not assign children to a group; children selected the group themselves. This further reduces the possibility that our pattern of results can be explained by demand characteristics.

- 3. Secondly, I am wondering if the authors could talk a little more about whether children are more quick to accept confirming evidence in the Group case, or is it that they are less likely to look for disconfirming evidence.**

Response: We thank Reviewer 1 for highlighting this interesting alternative. Please see our third response to Reviewer 2 who raises a similar point, but in summary: we ran an additional study (Study 2 in the revised manuscript) to address the different interpretations reviewers brought up regarding Study 1. Study 2 finds that children in the *Group Condition* were more convinced by the same set of group-supporting evidence. This cannot be explained by children avoiding disconfirming evidence in the *Group Condition* or children being more curious in the *No Group Condition*. Therefore, these results indicate children are quicker to accept confirming evidence.

- 4. I have a nitpicky point about the scale in Exp 2 [Study 3 in the revised manuscript]: it looks like a scale for how many more elephants vs. lions there are, rather than a confidence scale. I get what the authors were going for, but I wonder if children might have shared this confusion. If not, perhaps clarify what the training looked like.**

Response: We fully agree with Reviewer 1 that children's comprehension of the confidence scale is key to the current experiments. To ensure that children understood the scale in the intended way, we ran several manipulation checks. In Study 3, children were required to pass three comprehension check questions about the scale before continuing ("*where would you place your marker if you really thought there were more elephants?*", "*where would you place your marker if you really thought there were more lions?*", and "*and where would you place your marker if you didn't know?*"). In Study 2, children were required to pass four comprehension check questions ("*where would you place your marker if you were really sure there were more elephants?*", "*where would you place your marker if you mostly sure there were more elephants?*", "*where would you place your marker if you were kind of sure there were more elephants?*", and "*where would you place your marker if you weren't sure there were more elephants?*"). Taken together, the comprehension checks in Studies 2 and 3 provides strong evidence that children understood the purpose of the scale.

- 5. I'd also like the authors to consider some of the literature on testing counterintuitive claims - namely, the development of how they do so (older children are the ones who test counterintuitive claims, younger children tend to accept them), and integrate them into their discussion: Ronfard, S., Chen, E. E., & Harris, P. L. (2021) & Ronfard, S., Chen, E. E., & Harris, P. L. (2018).**

Response: We thank Reviewer 1 for highlighting these two relevant articles and have integrated them into the discussion on page 36.

RESPONSE TO REVIEWER 2:

- 1. This paper tests a neat idea that even children as young as four years old will hold on to a belief for social reasons - to affiliate with their group. This paper is the first to show biased belief formation and revision based on group identity at age four. I think on the surface it is both original and likely to be of wide interest. The paper is also well written and the conclusions are mostly (though see below for a minor quibble about experiment 1) supported by the evidence. I have a few points that I believe could be addressed in a revision, perhaps with more data.**

Response: We thank Reviewer 2 for their encouraging feedback on our manuscript.

- 2. I wish there were more age groups for comparison, and an adult comparison group, so that the authors could also explore potential developmental changes and mechanisms of change. For example, it seems to me that the initial belief is formed quickly because it exploits children's tendency to be credulous of in-group members... But older children (above age 6) may be less credulous initially than younger children, and more willing to trust themselves to look for evidence. Importantly, older children also have better probabilistic reasoning skills, and thus have a better grasp of the uncertainty involved about boxes that remain closed despite a seemingly deterministic pattern in the initial 3. Older children also are more patient, and can wait longer than 30 seconds for something they are interested in doing. All of this can potentially influence how far children will go to open boxes.**

Response: We agree with Reviewer 2 that additional age groups for comparison would be very interesting. We revised our discussion of how group membership may influence children's belief formation practices over a wide range of development on pages 35-36. The reason we selected children in the 4-6-year range is because earlier research indicates minimal group effects – such as children preferring their groups and being concerned with belonging to them – emerge at around 4 years of age and then may become stronger with age (Dunham, 2018).

Despite the strong reasons Reviewer 2 brings up regarding why older children might have more advanced epistemic practices (e.g., they are more patient and have heightened probabilistic reasoning skills), we find that both younger and older children have the necessary reasoning skills and are sufficiently patient—when they are in the *No Group Condition*. However, overall, we surprisingly find older children in Study 1 hold *less* accurate beliefs across conditions. One possibility is that older children were less curious. On average, older children opened fewer boxes than younger children. This difference trended towards significance ($p < .08$), and it may have influenced more older children to hold the incorrect belief on the whole. This is consistent with previous research which has demonstrated that older children and adolescents have reduced

exploratory tendencies to form beliefs (Schulz et al., 2019; Liquin & Gopnik, 2022). This work suggests as children grow up, they may become less concerned with forming the most accurate beliefs, and instead, increasingly make generalizations based on the evidence they have already seen. Nevertheless, overall, we find that both older and young children exercise sophisticated epistemic practices in the *No Group Condition* compared to children in the *Group Condition*.

- 3. The simple description of [study 1's] result was not that children "adjust their standards of evidence in order to hold the same beliefs as their group" but that children in the group condition explored fewer boxes (were less curious) than children in the no group condition... If the basic result of Experiment 1 is simply that group testimony restricts curiosity, it straightforwardly follows from the exploratory play literature (e.g. Bonawitz, shafto, et al) - children explore less when adults present their knowledge confidently. Even though this is an extension of prior work, I agree that it is a very interesting finding the context of groups. However, I'm not sure it is sufficient to warrant publication in this journal, at least not without a more complete picture of how much this result actually represents "political" reasoning as opposed to a social learning finding that follows from prior work.**

Response: We thank Reviewer 2 for making this insightful distinction. We collected additional data to respond to this point. It's true that in Study 1, children who belonged to a group may not necessarily have opened less boxes because they were adjusting their standards of evidence. Instead, our results could be explained by children in the *No Group Condition* being more curious or children in the *Group Condition* avoiding counterevidence in the later boxes (as Reviewers 1 and 3 mentioned). We view these possibilities as not mutually exclusive and think they all likely influence children's belief formation.

Nevertheless, to address Reviewer 2's point and the question of whether children are indeed adjusting their standards of (confirmatory) evidence, we ran an additional study (Study 2). Here, we kept the evidence children saw constant and measured their confidence in what they believed. We found children were more convinced by evidence if it supported a group belief compared to an unaffiliated group's belief. This pattern of results cannot be explained by children in the *No Group Condition* being more curious or children in the *Group Condition* avoiding counterevidence. Therefore, this suggests that children adjusted their standards of evidence in response to group-supporting evidence, which led children to open fewer pieces of evidence to arrive at a conclusion in Study 1.

In Study 3, we also found children adjusted their standards of evidence in response to group-opposing evidence — children were less convinced by seeing the same set of evidence if it opposed a group belief compared to an unaffiliated group's belief. Together, these three studies show group membership has a powerful influence on how children evaluate and weigh evidence, which in turn leads them to hold incorrect group beliefs.

- 4. The two groups' testimony is also a form of evidence... [the present studies] could be interpreted through this social learning lense as a finding that testimony from groups weights heavier than one's own observations. This could be rational for**

children to have evidential standards for testimony which weight it higher (e.g. Bridgers et al, also Sobel & Kushnir).

Response: We appreciate Reviewer 2's point, but do not see this as a problem given that children in both conditions—the *Group Condition* and the *No Group Condition*—receive testimony. The only difference across conditions is that in the *Group Condition*, children join one of the groups. We were interested in if belonging to a group and hearing their group's belief would in turn influence children's evidential standards, relative to children who hear the same testimony, but are not part of either group. Moreover, and importantly, we know from Study 3 that children do not directly weigh their group's testimony as stronger than an unaffiliated group's testimony—children's initial confidence across conditions was identical following exposure to the two group's beliefs.

RESPONSE TO REVIEWER 3:

- 1. This ms. has very significant strengths. It tackles an important issue, it's well written, and the experiments are very clever.**

Response: We thank Reviewer 3 for their positive feedback.

- 2. The glaring weakness is that the Authors have completely overlooked several very relevant (and huge!) strands of literature... There's a massive literature on how children evaluate information communicated by others (the literature on trust in testimony, which is very much about belief formation in a social context)... There's also a literature on peer pressure in children that age, for instance: Haun, D. B., & Tomasello, M. (2011). While I still believe the present ms. makes a contribution, the Authors have to situate their findings in the relevant literature so we can actually gauge its novelty, and they avoid overselling it. The Haun and Tomasello paper also raises the question of how much the children would actually ground their actions in their stated, group-consistent beliefs (something the Authors might discuss, but don't have to address with new data).**

Response: We thank Reviewer 3 for helping us clarify our theoretical contribution. We have revised our review of prior literature on children's social belief formation and tendency to conform on pages 4 and 5 of the introduction. For example, we have added a review of the trust in testimony literature Reviewer 3 mentions. The present studies mainly contrast with this literature in that here, we measure how social influences bias children's epistemic practices, including how children seek out and weigh evidence.

Additionally, we have added a discussion of how our paper relates to Haun & Tomasello (2011) on children's conformity. Please see our second reply to Reviewer 1 above for a full response on this topic, but briefly, the current manuscript differs from Haun & Tomasello in two important ways. First, unlike in Haun & Tomasello, children in our studies belong to a group, and we test how this influences their evaluation of evidence. Second, Haun & Tomasello show children will change their public expressions

due to social influences, not their actual beliefs. In the current manuscript, we show that group influences change children's beliefs through children's biased evaluation of evidence.

As Reviewer 3 mentions, it would be interesting to test if children changed their actions in line with their group beliefs. Corriveau & Harris 2010, who also investigate whether children defer to majority judgments (similar to Haun and Tomasello), find that children do not base their practical decision-making on their publicly stated expression. Future research could further test for this in the context of our experimental setup, including by having children fill out what they think is in the rest of the boxes; by having children place costly bets on the content of the boxes; or by telling children that if they get the correct answer, they can win a prize, for instance.

- 3. A more minor issue is that the Authors misinterpret the lack of evidence for an interaction between condition and age as evidence of absence (e.g. "Interestingly, 4-6-year-olds' epistemic practices were equally influenced by group membership."). They don't have a large enough sample to confidently conclude from the lack of a significant interaction that there is no interaction. In addition, the p values of most of these interactions are consistent trends in the direction of an interaction, suggesting that it's quite likely that there is an interaction. Maybe being able to look at the data (a graph in ESM?) might help get a better sense of what's going on.**

Response: In response to Reviewer 3's comment, we revised how we interpret the lack of an interaction effect in the manuscript on page 35 and added condition*age interaction graphs in the ESM. From the graphs, there is not a clear age trend across conditions. However, as Reviewer 3 points out, some p-values were consistent in the direction of an interaction, so it remains a possibility that we are capturing a trend, such that the effect of group membership on children's belief gets stronger with age.

- 4. When the Authors report the effects of the manipulation check, they might consider providing the data, and not just the inferential statistics, especially since some of the effects appear to be quite small.**

Response: We thank Reviewer 3 for this recommendation. We have included the means and standard deviations of the group manipulation check questions in the manuscript on pages 16, 21, and 30-31. We find a small but relatively consistent trend, such that children in the *Group Condition* tend to like and give more stickers to their group compared to children in the *No Group Condition*.

- 5. Would it be possible to run an (exploratory) analysis looking at whether condition has an effect, once you condition on the number of boxes opened in Experiment 1? I.e. could we know whether the effect of condition on the final belief is entirely due to the number of boxes opened? That seems relevant to understanding what the children are doing.**

Response: Reviewer 3 makes a great suggestion. In response, we ran a mediation analysis and found that the numbers of boxes children open explains the relationship between

condition and belief (please see pages 15, 32, and 35, as well as Figure 2C of the revised manuscript). In other words, the condition children are in (Group, No Group) determined the number of boxes children open, which in turn determined children's beliefs.

This nicely fits with the results of Study 3, such that group membership did not directly determine children's beliefs. Rather, group membership influenced how children evaluated evidence, which in turn influenced their beliefs. Together, this again suggests belonging to a group influences our beliefs indirectly through our evaluation of evidence.

- 6. If feels a bit like cheating to test whether children tend to believe their group after having removed the children who don't believe their group. I'm not sure how that can be handled, but I don't think the Authors can just say that they removed the children that show their hypothesis wrong.**

Response: We understand Reviewer 3's concern. However, it's important to note that only 4 out of 34 children did not believe their group. The vast majority of children believed their group. Second, our main interest in Study 3 was not whether children believe their group, but how they evaluate evidence that goes against their group's belief and whether they revise their group beliefs. We could thus only test this hypothesis with children who believed their group—which is why we preregistered this exclusion criterion. Third, we see a consistent effect of group membership on children's evaluation of evidence not only in Study 3, but also in Studies 1 and 2 (where we did not exclude children who did not form a group belief). This suggests that the effect we observe in Study 3 is not driven by our pre-registered exclusion criteria.

- 7. “First, children may do so because they are epistemically motivated and think their groups are reliable, accurate sources of information.” [quote from the manuscript]. I don't think that interpretation is consistent with the findings the Authors present right before: if it were epistemic, the children should just believe their group more in all the situations they have a chance to, which they don't do. An alternative, compatible with the Authors' second interpretation, is that most children try to avoid antagonizing members of their group by disagreeing with them: they don't look for too much evidence that might show them wrong (in Experiment 1), and they don't outright say their group is wrong (in Experiment 2 [Study 3 of the revised manuscript]). But they don't put any special stock in their group's belief otherwise (as shown by the evidence quote above).**

Response: We thank Reviewer 3 for drawing our attention to our account of why group membership influences children's belief formation practices. As Reviewer 3 outlines, this alternative is complementary to our second, instrumental account, such that children are motivated to affiliate with their group. However, it is important to note that while children may not directly put special stock in their group's belief, their group's belief does influence how they weigh evidence, which in turn influences what children believe. To us, it remains an open possibility whether epistemic and/or instrumental motivations ultimately underly this pattern. Children may either adjust their standards of evidence because they place more epistemic weight on information coming from their group (even if indirectly through their evaluation of evidence), and/or because they want to align their

beliefs with their group in order to affiliate. We discuss these two possibilities on pages 37-38 of the revised manuscript.

RESPONSE TO REVIEWERS 4 & 5 (co-reviewed):

- 1. There is a lot to like about this paper. It addresses an important and timely issue, does so in an innovative but methodologically careful and sound way, and both the results and writing are crystal clear.**

Response: We thank Reviewers 4 & 5 for their supportive thoughts.

- 2. Is this paper of sufficiently broad interest and importance to warrant publication in Nature Communications? The topic is clearly relevant to a broad readership, and this work integrates several compelling literatures in developmental psychology and beyond. That said, more work could be done to motivate why the questions asked ought to be studied through a developmental lens.**

Response: We thank Reviewers 4 & 5 for their constructive suggestion. In response, we have revised our introduction (see page 3) to highlight why examining group membership on children’s belief formation and evidence evaluation is of great practical importance and of particular relevance for scholars across a wide range of disciplines, along the lines Reviewers 4 & 5 suggest. In particular, while adult’s beliefs have often been found to be resistant to counterevidence and can be ‘locked into place’ due to partisan biases (e.g., Flynn et al., 2017; Pretus et al., 2023), children are a unique population in that they are only beginning to form their core moral, political, and evidence-based beliefs about the world. If group membership already biases their epistemic practices at a young age (as we demonstrate in the current manuscript), this suggests that intervening in how we form beliefs early in life is a promising direction to reduce polarization and foster intellectual humility.

- 3. Is the conceptual framing of the paper coherent and sound, and does it adequately motivate these particular studies? The conceptual framework lays out two broad sets of motivations that might underlie belief formation: epistemic motives and motives termed both “instrumental” and “social” in different parts of the paper. This framing is compelling, but it seems somewhat oversimplified, and potentially incorrect... It might be worth softening this dichotomy or even considering removing it altogether... The paper claims that children are at least in part driven by social [instrumental] over epistemic motives. But there is no direct evidence of this.**

Response: In hindsight, we agree with Reviewers 4 & 5 that our conceptual framing of epistemic versus instrumental motivations was oversimplified and did not fit the pattern of results. As Reviewers 4 & 5 point out, it is not clear exactly how instrumental motivations and epistemic motivations interact and result in belief formation. Further, it’s

not clear that the motivation to believe one's group is explained by instrumental motivations in the first place (as we describe in the discussion section). Alternatively, children could see their group as a reliable, accurate source of information, and thus their motivation to believe them could be purely epistemic (Levy, 2019). Because of these uncertainties, we have removed this dichotomy from the introduction and discussion, following Reviewers 4 & 5 suggestion. Instead, we highlight that group contexts may give rise to both epistemic and instrumental motivations, and then discuss how one would adjudicate between the two in the discussion section on pages 37-38.

- 4. Is the interpretation of the results sound, and might there be alternative explanations that ought to be tested or discussed? The interpretation of Study 2 [Study 3 in the revised manuscript] is a bit unclear. The group vs. no group effect in terms of the degree of belief revision is clear. The paper characterizes this as children in the group condition being resistant to counterevidence. But this may not be accurate. Children seem to move from being sure their group is right to completely unsure either way. That in itself is a belief revision that is consistent with the evidence they've seen, especially if they take their group-stated beliefs as equivalent to evidence (which they may, given that it is essentially testimony). Thus, a more accurate interpretation might be something like children's confidence in the ingroup-held belief is more deeply held than confidence in a belief stated by a group with whom they have no affiliation. But this is not the same as what the paper states.**

Response: We fully agree with Reviewers 4 & 5 and do not think that children in the *Group Condition* are resistant to counterevidence, just that they are more resistant to counterevidence, and tend to revise their beliefs less often than children in the *No Group Condition*. We revised the relevant sentence in the study-discussion of Study 3 on page 31.

More specific comments:

- 5. p. 10, line 261: It seems that the ingroup liking manipulation check in Study 1 wasn't significant but the resource allocation measure was. It's unclear what to make of this, or how it bears on (or does not bear on) the results.**

Response: Across Studies 1, 2, and 3 we find a strong effect of group membership. In Study 1, children who belonged to a group opened significantly less boxes ($p < .01$) and were significantly more likely to hold the incorrect belief ($p < .01$) in our main belief measures. In our preference measures, they trended towards liking their group more ($p = .05$) and gave them significantly more stickers ($p < .01$).

In Study 2, children who belonged to a group were significantly more confident in group-supporting evidence ($p < .01$) in our main belief measure. In our preference measures, they liked their group significantly more ($p < .01$) but did not give them significantly more stickers ($p = .18$).

In Study 3, children who belonged to a group were significantly less confident in group opposing evidence ($p = .02$) and were significantly less likely to revise their belief

($p < .01$) in our main belief measures. In our preference measures, they liked their group significantly more ($p < .01$) and gave them significantly more stickers ($p < .01$).

Taken together, we find a robust influence of group membership, particularly for our epistemic measures. One reason we did not always observe an effect for the liking and sharing ratings may be that our power analyses were based on our epistemic measure (our primary interest), not the preference measures. Nevertheless, in all studies, children significantly preferred their group on at least one of the two measures. These results are in line with previous work (e.g., Dunham, 2018), which indicate children generally like their ingroup more and share more resources with them relative to outgroup members. Here, we also find they were significantly influenced by group membership in their evaluation of evidence and belief formation practices.

6. p. 12, line 309: Why would older children have more inaccurate beliefs?

Response: One possibility is that older children were less curious. On average, older children opened fewer boxes than younger children. Although this difference was not significant ($p < .08$), it may have influenced more older children to hold the incorrect belief on the whole. This is consistent with previous research which has demonstrated that older children and adolescents have reduced exploratory tendencies to form beliefs (Schulz et al., 2019; Liquin & Gopnik, 2022). This work suggests as children grow up, they may become less concerned with forming the most accurate beliefs, and instead, increasingly make generalizations based on the evidence they have already seen. We have clarified this in more detail on pages 35-36 of the discussion.

7. p. 15, line 393: We think this should read “Step 5: Counterevidence” and a few lines down, “Step 6: Belief Measure 2” to be in-line with Figure 3.

Response: We thank Reviewers 4 & 5 for pointing out these typos. We have corrected them in the manuscript.

8. Would we anticipate gender effects? More discussion would be helpful.

Response: We thank Reviewers 4 & 5 for raising this interesting question. In response, we ran exploratory analyses with gender included as a predictor for all main analyses. However, we did not find any gender effects (Study 1. Boxes: $p = .98$, Belief: $p = .57$; Study 2. Confidence: $p = .16$; Study 3. Difference-score: $p = .75$, Revise: $p = .45$). In a review of the literature on minimal group belonging in childhood, it seems gender effects are not often observed (e.g., Dunham et al., 2011).

9. Did the authors have any parental data (e.g., parent political orientation)? If not, it might be worth discussing how parental factors might or might not be relevant here.

Response: We do not collect parental data such as political orientation as part of our demographic forms. However, examining the impact of different parental households and varying cultural environments on children’s group biases is a fascinating and valuable future direction. For instance, how might parental political orientation, SES, regional

ideological division, and cultural values (e.g., autonomy versus social harmony) impact children's group biases and belief formation practices? We hope to tackle these important questions going forward.

RESPONSE TO REVIEWER 1:

- 1. I believe the authors have been very responsive to reviews and the additional study does help in clarifying the distinction between testimony restricting exploratory play and testimony changing the standards of evidence. The notable result is that group testimony changes one's standard of evidence for evaluating claims - I think we do not *yet* have a mechanism to explain why, but that is a tall order, and one reserved for future work!**

Response: We thank Reviewer 1 for their positive assessment of our revision.

RESPONSE TO REVIEWER 3:

- 1. The Authors have done a good job of integrating the reviewers' feedback, including running a new and interesting study. I am now even more positive inclined towards the ms.**

Response: We thank Reviewer 3 for their encouraging feedback.

- 2. The results are sometimes framed in terms of demonstrating children put too much weight on the information from the in-group. However, and except in S1 (more on this below), there are no normative benchmarks. So it's equally possible that children are not putting enough weight on the no-group information. For instance, in S2, the children in the no-group condition are told the group's belief, then they see the evidence supporting that belief, and in spite of that they do not favor the group's belief at all (they are at the midpoint in terms of confidence). That suggests that they are very (overly?) skeptical of information from non-in-groups, instead of showing that they accept overly easily information from in-groups. So the conclusion they draw from S2 ("These results suggest that children overweight the evidentiary strength of evidence when it supports an ingroup belief.") seems to be completely backwards: the children underweight the evidence (and the opinion) of non-in-group members.**

Response: We appreciate Reviewer 3's point and agree that our studies do not include normative benchmarks; our data are not designed to determine what children *should* do in these situations. Because of this, we do not mean to claim that children in the *Group Condition* put *too much* weight on information coming from their group—rather, that they put *more* weight on information coming from their group than children in the *No Group Condition* put weight on information coming from a non-affiliated group. We have revised the relevant sentence in Study 2's discussion on page 23.

However, there might have been a misunderstanding regarding the procedure, and, more specifically, children's beliefs in the *No Group Condition*. Importantly, before seeing evidence, children in both conditions hear conflicting beliefs from the two groups (so children are always exposed, in both conditions, to the diverging opinions of both groups; for details, please refer to page 19 of our procedure). After seeing evidence, as

Reviewer 3 notes, children in the *No Group Condition* were at the midpoint in terms of confidence on our scale (5 out of 9). This translates to being between ‘kind of sure’ and ‘mostly sure.’ This suggests children are not overly skeptical of information coming from one of the non-in-groups, instead, they integrate the evidence they saw to arrive at a moderate judgement of confidence. Children in the *Group Condition*—who saw the same pieces of evidence—were two units more confident on our scale (7 out of 9), which translates to being ‘mostly sure.’

We believe that children in both conditions are *overly confident* in their beliefs (especially children in the *Group Condition*). All children saw only the contents of 2 out of 10 boxes. This is a limited sample that does not provide enough evidence of the overall distribution (i.e., the following 8 boxes could have the opposite animal as in the first 2). Additionally, all children heard a second group claim they believe there were more of the opposite animal as in the first group. Nevertheless, despite this uncertainty, children in the *Group Condition* were mostly sure in their belief and children in the *No Group Condition* were between mostly sure and kind of sure. This supports the notion that children who belong to a group are more readily inclined to accept information aligned with their group’s position.

- 3. There might be something I’m missing, but I don’t understand why the initial results from the no-group condition of S2 and S3 are so different. If I get it right, in S2, the children are told this group believes it’s mostly elephants, then they are showed two elephants. With all this information, they still have no confidence at all that there are more elephants. By contrast, in S3, the children are only told of the group’s belief, not shown any evidence, and they are nearly at ceiling in terms of believing that there are more elephants (for that matter, their belief seems to be higher even than their belief in the in group after the evidence has been shown in S2). What’s going on? Why would showing two elephants lower the belief that there are elephants? Did it draw the children’s attention to all the unopened boxes?**

Response: We thank Reviewer 3 for pointing out this comparison between Study 2 and Study 3. We believe there are two misunderstandings that may help resolve these concerns. First, as noted above, children’s confidence in the *No Group Condition* in Study 2 was in the middle of our scale (5 out of 9), indicating moderate confidence that there are more elephants. Second, children’s confidence in the *No Group Condition* in Study 3 was also in the middle of the relevant side of the scale (2.6 out of 5) before exposure to evidence, suggesting a similar moderate level of confidence. We have clarified the relevant sections and figures in the manuscript and apologize for any confusion caused by our presentation of the results.

Nevertheless, Reviewer 3 highlights an interesting question: why was children’s confidence in Study 2 (after seeing evidence) similar to children’s confidence in Study 3 (before seeing evidence) in the *No Group Condition*? We agree with Reviewer 3’s suggestion that only seeing two boxes opened in Study 2 may have drawn children’s attention to the large number of unopened boxes (8 out of 10 boxes remained unopened). Indeed, as noted earlier, viewing only 2 out of 10 boxes is a limited sample to arrive at a belief. In other words, this suggests that the presentation of limited evidence in Study 2

may have highlighted the uncertainty of the broader sample, ultimately restricting children's certainty in their belief.

- 4. Finally, my main issue is with the framing of the conclusion. The Authors write for instance “When children belong to a social group, they adjust their standards of evidence in order to hold the same beliefs as their group.” I don’t think the evidence particularly supports this “in order to” interpretation. It could also be that children put more weight on information provided by their in-group (as suggested later on in the conclusion), or that they don’t put enough weight on information provided by a non-in-group (see above). By analogy, imagine that the no-group condition were compared to an expert condition, in which someone who has, say, looked into the boxes, gives their opinion. If children accepted that expert opinion, it would obviously not show that they are motivated to hold the same beliefs as the expert—it would simply reveal that they are motivated to hold accurate beliefs (and that they trust the expert). If the children merely wanted to have the same belief as their in-group, they would have much easier ways of going about it (such as accepting the belief and not looking for any extra evidence). So I think the interpretation can’t be that straightforward. This issue recurs throughout the discussion, and it really has to be fixed—the results just don’t warrant that conclusion.**

Response: We agree with Reviewer 3 regarding these comments on the framing of our conclusion. While our findings indicate that children are more likely to align their beliefs with their in-group compared to a non-ingroup, our data do not allow us to determine whether this pattern is driven by epistemic motivations (seeking accurate beliefs) or instrumental motivations (affiliating with their group). We elaborate on this point in the discussion section on pages 36-37. We have also revised the framing throughout the discussion, including removing the ‘in order to’ phrasing. For example, on page 34, we have replaced the original statement, “*the process of adjusting one’s epistemic practices in order to hold group beliefs has deep developmental roots*” with the revised phrasing, “*the process of adjusting one’s epistemic practices in group contexts has deep developmental roots.*” We made similar adjustments elsewhere (e.g., pages 31, 33, 35, 36). We thank Reviewer 3 for helping us refine these points.

Minor comments:

- 5. In the introduction, the Authors should be clearer about their theoretical framework: they go from belief formation in general to “how children evaluate evidence to form beliefs,” which is a more specialized literature, back to belief formation more generally in the case of trust in testimony without explaining the differences between the objects of these literatures. So it’s great that the Authors have added the literature that had been omitted the first time around, but it feels like it’s been added maybe a bit thoughtlessly, without proper integration.**

Response: In response to this point, we have clarified the distinctions between children’s evaluation of evidence and the role of social testimony in belief formation in the introduction on pages 4-5. Specifically, we now more explicitly differentiate between

studies on how children evaluate firsthand evidence and studies on how social testimony influences belief formation.

- 6. I think the Authors are overselling the potential of their findings for interventions that would have any type of effect in adulthood. To bolster their argument, they should mention studies that show such interventions are effective, otherwise I think it's too speculative (and the paper doesn't need that to be interesting).**

Response: We did not intend to overstate the potential for childhood interventions on group biases, but rather to highlight that early interventions may be beneficial given the limited success of adult interventions. Adult's beliefs are notoriously resistant to evidence and accuracy-based interventions. To our knowledge, similar interventions have not been tested in children, making this an open and important question for future research. To address this concern, we have softened the language in the introduction on page 3 and in the discussion on page 38.

In response to Reviewer 3's comment, we also added references in the discussion on page 38 to support the view that intergroup interventions during childhood and adolescence may be especially impactful. For example, Wölfer et al., (2016) found that intergroup contact in adolescence was associated with more positive intergroup attitudes over time, whereas similar effects were not observed in older cohorts. Their analysis indicated that adolescence may be a particularly sensitive period for shaping intergroup attitudes. Similarly, Rutland & Killen (2015) argue that interventions may be most effective before biases are fully formed in adulthood. In other words, early interventions have the potential to address the formation of group biases or reframe them before they solidify.

- 7. "epistemic weight" just weight (there are I believe a couple other instances of unnecessary 'epistemic')**

Response: We have removed 'epistemic' from the phrase 'epistemic weight' on page 17 and from 'epistemic source' on page 5.

- 8. The figure legend should indicate which study they refer to.**

Response: We have updated the figure legends to explicitly indicate which study each figure corresponds to.

- 9. I'm not a stat expert, but it would be nice to have a sense of why the power analysis yielded a required sample size in S2 twice as large as in S3.**

Response: Our power analyses for both studies were based on pilot data, which yielded the respective sample sizes. Differences in the estimated variability or effect sizes between the pilot datasets likely contributed to the discrepancy between Study 2 and 3. We have clarified the nature of our power analyses on pages 18 and 24.

- 10. The main effect of age in S1 could also be explained by older children getting bored more easily by this game (waiting to look into boxes to find toy animals), so I don't think much stock should be put on this.**

Response: We agree with Reviewer 3 that the main effect of age in Study 1 could be influenced by older children becoming less engaged with the task. To not overstate this result, we have noted this alternative explanation in the revised manuscript on page 35.

- 11. I couldn't find the pre-registration for the new study (there are only two pre-registrations in the link provided).**

Response: We apologize for this error. We have now ensured that all pre-registrations are available in the links provided on page 8.

RESPONSE TO REVIEWERS 4 & 5 (co-reviewed):

- 1. Our review of the initial submission included several broader questions: (1) Is the paper of sufficiently broad interest and importance to warrant publication in this journal? (2) Is the conceptual framing of the paper coherent and sound, and does it adequately motivate these particular studies? and (3) Is the interpretation of the results sound, and might there be alternative explanations that ought to be tested or discussed? Having carefully re-read the revised manuscript and response letter, we believe that our initial reviews were appropriately addressed and think that this revision will make a strong contribution to the literature.**

Response: We thank Reviewers 4 & 5 for their positive feedback!

- 2. In addition to assessing the revision in light of our own reviews, we were asked to comment on whether Reviewer 2's reviews were appropriately addressed. In general, we believe that the revised manuscript addresses Reviewer 2's concerns, with two possible exceptions, which we discuss below.**

Response: We thank Reviewers 4 & 5 for taking the time to assess our responses to Reviewer 2.

- 3. First, Reviewer 2's Comment 2 asks about the possibility of testing additional age groups. This comment is addressed in part with the existing data, but it was unclear to us to what age group both the reviewer and response were referring when referencing "older children." It seems as though Reviewer 2 was talking about children over the age of 6, which—to our knowledge—the revision cannot currently speak to, even with the new data. Given this comment, it was surprising to us that the revision didn't include a slightly older age range in the new collected sample. Despite this, we do think that the revision is strong enough to warrant publication without new data examining additional age groups. The authors might consider justifying in a bit more detail why they focused on their young age range, and/or**

add to the General Discussion why (or why not) examining a broader age range could be an important future direction. The authors do this to some extent in the current General Discussion, but this could be expanded substantially.

Response: We selected children in the 4-6-year range because prior research suggests that minimal group effects begin to emerge around this age (Dunham et al., 2011). We were interested in if the effects of group membership on children's epistemic practices would grow stronger or weaker in this developmental window (e.g., Patterson & Bigler, 2006). We did not extend the age range in our newly collected sample for two key reasons. First, we were concerned that older children might not take the game and the group context as seriously, as it was designed for (and piloted with) 4-6-year-olds. This would make direct comparison across a wider range of development challenging. Second, our power analysis suggested a sample of 124 children for the new study. We decided it would not be feasible to expand the sample to include additional age groups. Nevertheless, we agree with Reviewer's 4 & 5 that testing additional age groups is an interesting future direction and have expanded upon this point in the discussion section on pages 35-36.

- 4. Second, we weren't entirely sure how to interpret Reviewer 2's Comment 4. Our read of this comment seemed to be more about whether one can pull apart the influence of the groups' testimony from children's observation of evidence, both of which could be forms of evidence that shape children's responses. Based on the authors' response and our reading of the manuscript, it seems this may have stemmed from a misunderstanding. Specifically, R2's Comment 4 seems to suggest they understood that children in the no group condition did not receive testimony from a group, but as the authors' response clarifies, the manipulation was not about the source of the testimony, but rather whether or not children belonged to one of the groups—which in the no group condition, they did not. Thus our sense is that this comment has been addressed sufficiently.**

Response: We appreciate Reviewer 4 & 5's consideration of this point. We are glad that our response helped resolve the potential misunderstanding.

- 5. The revision now includes an exploratory mediation in Study 1 (p. 15). We believe this is a cross-sectional mediation, but a lot of the language in Study 1 and the General Discussion surrounding this exploratory pattern sounds pretty causal—we would recommend softening the language, or more clearly articulating how causality can be inferred from this exploratory analysis.**

Response: In response to this comment, we have softened the language in Study 1 and the General Discussion to ensure that causal claims are not overstated.

- 6. Something about the combined presentation of Study 2's method across the figure and text made us initially a bit confused about when exactly the belief measure was administered. Upon a few re-reads, we assume it's after the evidence (two elephants) was provided. Also, in the figure, the belief measure (Step 5) and confidence in belief**

(DV) are the same thing, correct? Making the order of the measures clearer in text might allay potential confusion of future readers.

Response: We thank Reviewers 4 & 5 for drawing our attention to this potential source of confusion. The belief measure was indeed administered after children saw the two elephants, and Step 5 and our DV were the same thing. To clarify these points, we have revised Figure 3 and updated the text for Study 2's procedure.

- 7. At the end of Study 1's Introduction, the revision "predicted that this tendency would be stronger for older children than younger children (following the developmental trend in Roberts et al., 2021)" (p. 8). This "developmental trend" cited above wasn't really spelled out in the Introduction (unless we missed it), though it does appear in the General Discussion. Could this trend be more clearly articulated before the revision delves into the studies?**

Response: We have more clearly spelled out the development trend of Roberts et al., (2021) in the introduction on pages 7-8.

- 8. Below we detail what we think are several typos:**
- a. Second-to-last line of Abstract: children' → children's**
 - b. Last line of p. 7 says that children were assigned to one of two groups. This is incorrect, right? Children chose one of two groups across studies?**
 - c. Last line of p. 17: Group Condition → No Group Condition (right?)**

Response: We thank Reviewers 4 & 5 for pointing out these typos—we have fixed them.

RESPONSE TO REVIEWER 3:

- 1. Thank you for your work on the revisions, I think it's considerably strengthened the ms.**

Response: We thank Reviewer 3 for their positive assessment of the revision and for their valuable role in strengthening the manuscript.

RESPONSE TO REVIEWER 4 & 5:

- 1. Thank you for the opportunity to review this revision. We believe that our reviews were appropriately addressed and think that this paper will make a strong contribution to the literature.**

Response: We're grateful to Reviewers 4 & 5 for their thoughtful feedback and constructive comments, which meaningfully improved the manuscript.

- 2. Regarding the editor's specific requests, we don't have any major concerns about the discrepancy between the power analysis for Studies 2 and 3. That said, unless we missed it, there also isn't a lot of information in the manuscript to assess this discrepancy. For Study 2, the paper states, "We aimed to include 124 participants in our final sample, determined by a power analysis based on pilot data. The analysis suggested we would have an average power of $1-\beta > 0.80$ to detect a significant effect of condition." For Study 3, the paper states, "For Study 3, We aimed to include 60 participants in our final sample, determined by a power analysis based on pilot data. The analysis suggested we would have an average power of $1-\beta > 0.80$ to detect a significant effect of condition." Both studies seem to have primary DVs on comparable scales—perhaps Study 3 was powered with a larger effect in mind? The review response document states that "Our power analyses for both studies were based on pilot data, which yielded the respective sample sizes. Differences in the estimated variability or effect sizes between the pilot datasets likely contributed to the discrepancy between Study 2 and 3." This makes sense to us—the authors could consider adding more information about their power analyses (e.g., the effect sizes they were powering for) into the supplement or main text for further clarity.**

Response: We appreciate Reviewers 4 and 5's suggestion. The observed effect sizes in our pilot studies differed, which informed the respective sample size targets: Study 2's pilot yielded an f^2 of 0.13, corresponding to a small-to-medium effect, while Study 3's pilot yielded a Cohen's d of 0.79, reflecting a large effect. We have added this information to the Methods section for Study 2 (pg. 26) and Study 3 (pg. 28).